# GNN-AS-JUDGE: UNLEASHING THE POWER OF LLMS FOR GRAPH LEARNING WITH GNN FEEDBACK

**Ruiyao Xu, Kaize Ding**
Department of Statistics and Data Science
Northwestern University
`ruiyaoxu2028@u.northwestern.edu`
`kaize.ding@northwestern.edu`

## ABSTRACT

Large Language Models (LLMs) have shown strong performance on text-attributed graphs (TAGs) due to their superior semantic understanding ability on textual node features. However, their effectiveness as predictors in the low-resource setting, where labeled nodes are severely limited and scarce, remains constrained since fine-tuning LLMs usually requires sufficient labeled data, especially when the TAG shows complex structural patterns. In essence, this paper targets two key challenges: (i) the difficulty of generating and selecting reliable pseudo labels on TAGs for LLMs, and (ii) the need to mitigate potential label noise when fine-tuning LLMs with pseudo labels. To counter the challenges, we propose a new framework, *GNN-as-Judge*, which can unleash the power of LLMs for few-shot semi-supervised learning on TAGs by incorporating the structural inductive bias of Graph Neural Networks (GNNs). Specifically, *GNN-as-Judge* introduces a collaborative pseudo-labeling strategy that first identifies the most influenced unlabeled nodes from labeled nodes, then exploits both the agreement and disagreement patterns between LLMs and GNNs to generate reliable labels. Furthermore, we develop a weakly-supervised LLM fine-tuning algorithm that can distill the knowledge from informative pseudo labels while mitigating the potential label noise. Experiments on multiple TAG datasets demonstrate that *GNN-as-Judge* significantly outperforms existing methods, particularly in low-resource regimes where labeled data are scarce.[1]

## 1 INTRODUCTION

Text-Attributed Graphs (TAGs), where nodes correspond to text documents and edges represent their relationships, are pervasive across various applications such as citation networks, social media platforms, and e-commerce ecosystems (Chang & Blei, 2009; Yang et al., 2015; Hu et al., 2020; Wu et al., 2025). Unlike conventional attributed graphs, TAGs encode raw textual contents rather than numerical values, which requires more dedicated mechanisms to effectively capture semantic information while preserving structural relationships. Recent advances in Large Language Models (LLMs) have shown exceptional capabilities in text understanding (Brown et al., 2020; Dong et al., 2022; Minaee et al., 2024), driving growing interest in leveraging their exceptional text understanding capabilities to address TAG-related tasks (Chen et al., 2024b; Tang et al., 2024; Ye et al., 2024; Chen et al., 2024c; Hu et al., 2024; Wang et al., 2025; Wu et al., 2025). Many previous studies investigate *LLM-as-Predictors* approaches, which employ LLMs as direct predictors by integrating graph context through graph encoders or carefully crafted prompts (Tang et al., 2024; Chen et al., 2024a;b).

It is noteworthy that current research of *LLMs-as-Predictors* for node classification on TAGs, primarily focuses on the *supervised setting* where abundant labeled data is accessible. This is mainly because LLMs lack GNNs' message passing mechanisms to leverage unlabeled nodes, and thus require sufficient supervision signals for effective fine-tuning (Chen et al., 2024a; Tang et al., 2024; Ye et al., 2024). However, real-world graphs are usually sparsely labeled, and thus directly applying existing methods to such *few-shot semi-supervised setting* (Ding et al., 2020; Wan et al., 2021; Ding et al.,

---

[1]Code is available at https://github.com/rux001/GNN-as-Judge.

2022; Yu et al., 2025) will easily lead to overfitting and poor generalization (Wu et al., 2025; Tang et al., 2024; Ye et al., 2024). Although one can leverage pseudo-labeling techniques to augment the limited labeled training data (Lee et al., 2013; Rizve et al., 2021; Qiao et al., 2018; Liu et al., 2022; Sun et al., 2020; Li et al., 2018), previous studies have recognized that "easy" pseudo labels with high confidence provide limited learning signal, whereas "hard" samples are more informative but introduce greater label noise (Bengio et al., 2009; Kumar et al., 2010; Mukherjee & Awadallah, 2020). This challenge becomes even more pronounced when incorporating LLMs, leading to the following two unresolved challenges in few-shot semi-supervised node classification on TAGs:

- ❶ *How to go beyond the knowledge of LLMs to obtain reliable pseudo-labeled data?* Since LLMs are inherently difficult to interpret complex graph structural patterns (Huang et al., 2024; Guo et al., 2023), solely relying on LLMs with no structural inductive bias to generate pseudo labels is less desirable. More critically, not all unlabeled nodes are equally valuable for pseudo-labeling, thus, selecting the most influential subset of unlabeled nodes is crucial for maximizing performance under computational constraints. Although text-based serialization methods try to encode structural context into the prompts, those methods could still struggle with self-generated pseudo labels due to the hallucination and self-bias of LLMs (Chen et al., 2024b; Luo et al., 2024; Liu et al., 2022; Gao et al., 2024). Generating reliable pseudo-labeled examples that encode not only textual but also structural inductive biases is therefore crucial for enabling LLMs to transcend their inherent knowledge limitations on graphs.

- ❷ *How to extract the knowledge from pseudo-labeled data while mitigating the potential label noise during LLM fine-tuning?* Despite pseudo-labeling showing its empirical effectiveness in many fields (Lee et al., 2013; Rizve et al., 2021), the potential label noise has been a longstanding problem. Especially for those "hard" pseudo-labeled examples that are more valuable for training the model, they could also bring more label noise if the labels are incorrect. Simply performing LLM supervised fine-tuning with the noisy pseudo labels can lead to performance degradation (Shumailov et al., 2023; Kim et al., 2023; Cheng et al., 2020), which necessitates the need to develop a new learning algorithm that can effectively distill the knowledge and mitigate the label noise when fine-tuning with pseudo-labeled data.

In this paper, we propose *GNN-as-Judge*, a novel framework that fine-tunes LLMs on sparsely labeled graphs using feedback from GNNs. Instead of mining the "easy" and "hard" pseudo-labeled nodes solely based on LLM itself like standard self-training approaches (Ma et al., 2017; Lee et al., 2013; Mukherjee & Awadallah, 2020), at its core, a GNN with structural inductive bias acts as a judge to provide additional guidance for LLM to generate reliable pseudo-labels. *GNN-as-Judge* strategically leverages both agreement and disagreement between GNN and LLM as signals to identify not only "easy", but more remarkably, "hard" pseudo labels that LLMs are more likely to make mistakes. To further mitigate the potential label noise, especially in the harder examples, we develop a weakly-supervised LLM fine-tuning algorithm that jointly performs fine-tuning on the two selected pseudo-labeled node sets. Specifically, in addition to applying supervised LLM instruction tuning on the agreement (easy) node set, we propose to conduct preference tuning on the disagreement (hard) node set, which allows LLM to learn relative preferences between predictions from the two models. To summarize, our contributions are mainly three-folds:

- We study the problem of *LLMs-as-Predictors* for graph few-shot semi-supervised learning, a fundamental but underexplored research problem, where the key challenges lie in selecting reliable pseudo labels and mitigating label noise during fine-tuning.

- We propose *GNN-as-Judge*, a novel framework that positions GNNs as judges to select reliable pseudo labels for fine-tuning LLMs. *GNN-as-Judge* is also equipped with a new weakly-supervised fine-tuning algorithm that can further mitigate label noise during LLM fine-tuning.

- We conduct comprehensive experiments on different TAG datasets with various scales. Results demonstrate that *GNN-as-Judge* significantly outperforms both traditional GNN-based approaches and other LLM-based baselines, especially in extreme low-resource scenarios.

## 2 RELATED WORK

**LLMs for Graph Learning.** Recent research has extensively explored applying (L)LMs to TAGs, yielding significant advancements in feature encoding, node classification, and link prediction (Chen et al., 2024b; Tang et al., 2024; Ye et al., 2024; Hu et al., 2024; Wang et al., 2025; Wu et al., 2025). Early work primarily employed small-scale pretrained language models such as BERT (Devlin et al.,

2019) or RoBERTa (Liu et al., 2019) as text feature encoders to extract enhanced representations for graph learning (Yang et al., 2021; Li et al., 2023; Wen & Fang, 2023; Zhao et al., 2023). With the advent of powerful large language models such as ChatGPT (Achiam et al., 2023), researchers began utilizing these models in two primary ways: *as-enhancers* and *as-predictors*. *LLM-as-Enhancers* methods (He et al., 2024; Chen et al., 2024c; Sheng et al., 2025; Yu et al., 2025; Liu et al., 2024; Wang et al., 2025) utilizes LLMs to generate enhanced knowledge in the form of explanations, embeddings or labels based on the original graph data. For example, TAPE (He et al., 2024) uses LLM-generated explanations as enriched node features for GNN learning. *LLM-as-Predictors* methods frames graph problems as text generation tasks, directly utilizing LLMs as predictors to output classification or prediction results by transforming graph structures into natural language descriptions (Wang et al., 2024a; Ye et al., 2024; Chen et al., 2024a; Tang et al., 2024; Wang et al., 2024b; Chen et al., 2024b; Hu et al., 2024). For instance, LLaGA (Chen et al., 2024a) introduces template-based graph-to-text conversion with a specialized projector for structure comprehension, while GraphGPT (Tang et al., 2024) performs multi-stage instruction tuning to align LLMs with graph patterns. Although *LLM-as-Predictors* methods show promising performance and zero-shot transfer ability, they typically assume abundant labeled nodes are available. Recent work (Wu et al., 2025) demonstrates that standalone LLMs are weak predictors with limited labeled data, and incorporating graph structure is essential for achieving satisfactory performance.

**Pseudo Label Selection.** In semi-supervised learning, pseudo-labeling (Lee et al., 2013; Shi et al., 2018) serves as an effective solution by augmenting limited labeled datasets with generated labels. To mitigate the potential label noise, previous research usually leverages model's confidence to select "easy" samples that are considered to be clean (Ma et al., 2017; Kumar et al., 2010). Recent research, however, argues that there is little information to gain with these "easy" examples (Mukherjee & Awadallah, 2020) and merely relying on high-confidence examples may make the model self-biased (Rizve et al., 2021; Liu et al., 2022). Consequently, current research emphasizes the importance of mining both "easy" and "hard" samples during training to maximize model performance (Liu et al., 2022; Mukherjee & Awadallah, 2020; Shrivastava et al., 2016). Nevertheless, identifying the "easiness" or "hardness" of samples is often non-trivial,especially for LLMs.

## 3 PROPOSED APPROACH

In this section, we propose an LLM-GNN pseudo co-labeling framework *GNN-as-Judge* that addresses LLM pseudo-labeling challenges through three core designs as shown in Figure 1: (i) a subset selection strategy that identifies nodes with most information from labeled nodes for pseudo labeling; (ii) a collaborative pseudo label selection mechanism that leverages GNN as complementary signal sources for LLM to generate high-quality pseudo labels, and (iii) a weakly-supervised fine-tuning algorithm that mitigates label noise when fine-tuning LLM with generated pseudo labels.

### 3.1 PRELIMINARIES

We consider the problem of *few-shot semi-supervised node classification* on text-attributed graphs (TAGs) defined as $\mathcal{G} = (\mathcal{V}, \mathcal{E}, \mathbf{A}, \mathbf{X})$, where $\mathcal{V} = \{v_1, v_2, \ldots, v_N\}$ is the node set, $\mathcal{E} \subseteq \mathcal{V} \times \mathcal{V}$ is the edge set, $\mathbf{A} \in \{0, 1\}^{N \times N}$ is the adjacency matrix, and $\mathbf{X} \in \mathbb{R}^{N \times F}$ contains node features derived from text. Given a small labeled set $\mathcal{V}_{\text{train}}$ where each class $c \in \{1, \ldots, C\}$ has exactly $k$ labeled nodes, along with validation set $\mathcal{V}_{\text{val}}$, the goal is to predict labels for test nodes $\mathcal{V}_{\text{test}} = \mathcal{V} \setminus (\mathcal{V}_{\text{train}} \cup \mathcal{V}_{\text{val}})$. Common approaches include: (1) *GNN-as-Predictors* methods that learn node representations via message passing, yielding predictions $\hat{y}_i^{\text{GNN}} = f_\phi(v_i, \mathbf{A}, \mathbf{X})$ with trainable parameters $\phi$, and (2) *LLM-as-Predictors* methods that construct prompts $\mathcal{P}(v_i)$ from textual attributes and generate predictions $\hat{y}_i^{\text{LLM}} = \mathcal{M}_\theta(\mathcal{P}(v_i))$ using language models with parameters $\theta$.

### 3.2 GNN-AS-JUDGE FOR LLM PSEUDO LABELING ON GRAPHS

To break the bottleneck of using LLM-generated pseudo labels, our approach tries to mine "easy" and "hard" pseudo labeled nodes based on the agreement and disagreement between LLM and GNN, in order to provide reliable pseudo labels for fine-tuning the LLMs. To avoid the computational constraints of pseudo-labeling over the entire unlabeled set, we first identify the most influential unlabeled nodes according to graph structures, then apply our collaborative pseudo labeling framework

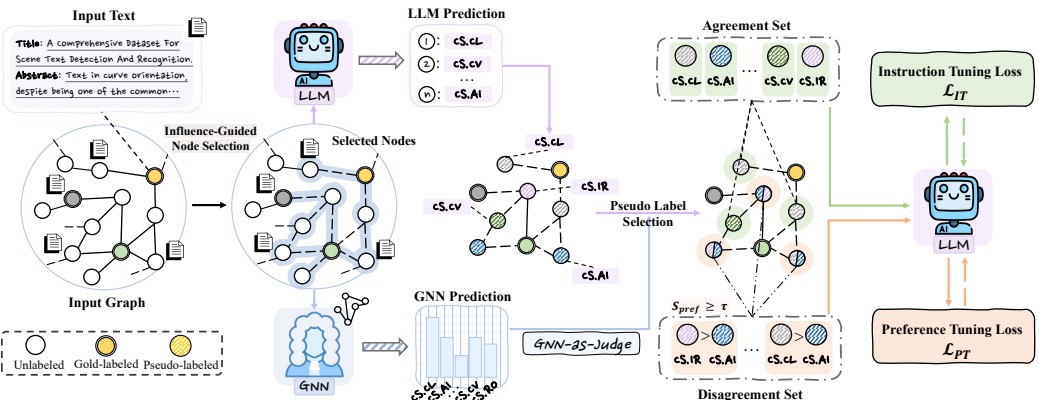

Figure 1: Framework of *GNN-as-Judge* for few-shot semi-supervised node classification on TAGs.

to this selected subset. Specifically, we leverage the complementary strengths of two distinct models: a structure-aware GNN $f_\phi$ and a text-centric LLM $\mathcal{M}_\theta$, both trained on the labeled set $\mathcal{V}_{\text{train}}$.

**Influence-Guided Node Selection for Pseudo Labeling.** Due to computational constraints and efficiency considerations, it is crucial to select a candidate node subset from the entire unlabeled node set for pseudo labeling. While LLMs excel at processing textual information, they lack awareness of graph structures and cannot readily capture the influence of labeled data on unlabeled nodes (Dai et al., 2025; Chen et al., 2024b). To address this, we employ GNNs as a structural proxy to identify subsets of nodes that carry the most information from labeled data. We leverage the concept of *node influence* that quantifies the extent to which one labeled node's representation can impact another unlabeled node's representation through the graph structure (Xu et al., 2018; Huang & Zitnik, 2020; Wang & Leskovec, 2020; Zhang et al., 2021; Wang et al., 2022). We first provide the formal definition:

**Definition 1** (**Node Influence**). *Node influence $I_{v_i, v_j}$ of node $v_i$ on node $v_j$ in the final GNN output is: $I_{v_i, v_j} = \left\| \partial \mathbf{x}_{v_j}^{(\infty)} / \partial \mathbf{x}_{v_i}^{(\infty)} \right\|$, where $\mathbf{x}_{v_i}^{(\infty)}$ and $\mathbf{x}_{v_j}^{(\infty)}$ are the final node representations of $v_i$ and $v_j$ learned by the neighborhood aggregation mechanism after infinite layers, the norm is any subordinate norm, and the Jacobian measures how a change in the representation of $v_i$ translates to a change in the representation of $v_j$.*

Definition 1 captures how effectively labeled nodes can propagate their information to unlabeled nodes. Nodes with high influence from labeled data are ideal candidates for pseudo labeling because: (i) they receive stronger signals from the labeled set, making pseudo labels more reliable, and (ii) they better reflect the distributional characteristics of the labeled nodes, ensuring representativeness and diversity. We then propose Theorem 1, which establishes that the influence of node $v_i$ on node $v_j$ decays with their distance and provides a computable upper bound for node influence. The proof is provided in Appendix D.1.

**Theorem 1.** *Let $\mathcal{P}_{v_i, v_j}$ denote the set of all paths between nodes $v_i$ and $v_j$, and let $\mathcal{P}_{v_i, v_j}^* \subseteq \mathcal{P}_{v_i, v_j}$ denote the set of shortest paths. For any path $t \in \mathcal{P}_{v_i, v_j}$, let $D_{GM}^t$ be the geometric mean of node degrees occurring on path $t$ defined as $D_{GM}^t = \left( \prod_{v_k \in t} D_{v_k v_k} \right)^{1/|t|}$, where $|t|$ is the path length and $D_{v_k v_k}$ is the degree of node $v_k$. Define $D_{GM}^* = \min_{t \in \mathcal{P}_{v_i, v_j}^*} \{ D_{GM}^t \}$ as the minimum geometric mean among shortest paths, $h^* = d(v_i, v_j)$ as the shortest path distance between $v_i$ and $v_j$, and $|\mathcal{P}_{v_i, v_j}^*|$ as the number of shortest paths between $v_i$ and $v_j$. Then, the node influence $I_{v_i, v_j}$ from $v_i$ to $v_j$ satisfies:*

$$I_{v_i, v_j} = \left\| \partial \mathbf{x}_{v_j}^{(\infty)} / \partial \mathbf{x}_{v_i}^{(\infty)} \right\| \leq \frac{|\mathcal{P}_{v_i, v_j}^*|}{(D_{GM}^*)^{h^*}} \tag{1}$$

Based on Theorem 1, we define the *influence score* of an unlabeled node $v_j \in \mathcal{V}_{\text{unlabeled}}$ as its maximum influence from any labeled node:

$$\mathcal{IS}(v_j) = \max_{v_i \in \mathcal{V}_{\text{train}}} I_{v_i, v_j} = \max_{v_i \in \mathcal{V}_{\text{train}}} \frac{|\mathcal{P}_{v_i, v_j}^*|}{(D_{GM}^*)^{h^*}} \tag{2}$$

We then rank all unlabeled nodes based on their influence scores and select the top $K$ nodes with the highest values:

$$\mathcal{V}_{\text{selected}} = \text{TopK}\left(\{\mathcal{IS}(v_j) : v_j \in \mathcal{V}_{\text{unlabeled}}\}, K\right) \tag{3}$$

This formulation ensures we select unlabeled nodes that are most strongly influenced by the labeled node set, maximizing the potential for effective information propagation during pseudo labeling. After obtaining the predictions from both the GNN and LLM on the selected node set $\mathcal{V}_{\text{selected}}$, we further partition the selected unlabeled node set into agreement node set $\mathcal{V}_{\text{agreed}} = \{v_i \in \mathcal{V}_{\text{selected}} \mid \hat{y}_i^{\text{GNN}} = \hat{y}_i^{\text{LLM}}\}$ and disagreement node set $\mathcal{V}_{\text{disagreed}} = \{v_i \in \mathcal{V}_{\text{selected}} \mid \hat{y}_i^{\text{GNN}} \neq \hat{y}_i^{\text{LLM}}\}$. The intuition behind using both agreement and disagreement node sets is that they serve complementary roles in effective LLM fine-tuning. While agreement nodes provide more reliable learning signals that consolidate the model's understanding, carefully selected disagreement nodes offer challenging examples that provide more informative learning signals.

**Agreement Node Set Selection with GNN Feedback.** We assume that nodes in the agreement set are more likely to have correct pseudo labels, as agreement between models with distinct inductive biases suggests higher label reliability. We provide theoretical justification for this assumption, which shows that the expected accuracy of the agreement set is strictly than that of either individual model:

**Theorem 2.** *Consider $\mathcal{V}_{selected}$ with ground truth labels $\mathbf{y}^*$. Let $p_{LLM}$ and $p_{GNN}$ denote the individual accuracies of the LLM and GNN respectively. Under conditional independence of model errors due to different inductive biases and uniform error distribution across incorrect classes, the accuracy of the agreement set satisfies:*

$$P(y_i^* | v_i \in \mathcal{V}_{agreed}) \geq \frac{p_{LLM} \cdot p_{GNN}}{p_{LLM} \cdot p_{GNN} + \frac{(1-p_{LLM})(1-p_{GNN})}{C-1}} \tag{4}$$

*where $C$ is the number of classes. Furthermore, this lower bound exceeds $\max(p_{LLM}, p_{GNN})$ when both models perform better than random guessing.*

The proof is provided in Appendix D.2. This theoretical result confirms that the agreement set provides higher-quality pseudo labels compared to using either model individually.

**Disagreement Node Set Selection with GNN Feedback.** While pseudo-labeled nodes in the agreement node set are provably more reliable, simply using easy, self-generated labels where the LLM already predicts correctly provides limited new learning signals and may lead to overfitting. To counter this issue, we propose to employ GNN as a judge for selecting high-quality pseudo labels in the disagreement node set. Since our method selects structurally influential nodes for pseudo labeling and the GNN's message-passing mechanism can exploit local neighborhood information that is inaccessible to the LLM, the GNN is assumed to be more reliable than LLM for pseudo labeling in this set. To further ensure the quality of pseudo labels, we utilize GNN's probability distribution over different classes as a natural indicator of its preference strength. For each node $v_i \in \mathcal{V}_{\text{disagreed}}$, we compute a preference score measuring how strongly the GNN favors its own prediction over the LLM's prediction: $S_{\text{pref}}(v_i) = P_{\text{GNN}}(\hat{y}_i^{\text{GNN}} \mid v_i) - P_{\text{GNN}}(\hat{y}_i^{\text{LLM}} \mid v_i)$, where $P_{\text{GNN}}(\hat{y}_i^{\text{GNN}} \mid v_i)$ is the GNN's predicted probability for its top class, and $P_{\text{GNN}}(\hat{y}_i^{\text{LLM}} \mid v_i)$ is the probability assigned to the LLM's predicted class. A larger preference score indicates a stronger conviction by the GNN in its own prediction relative to the LLM's alternative. We then select the final subset of nodes, $\mathcal{V}'_{\text{disagreed}}$, by retaining only those where the GNN's preference score exceeds a predefined confidence threshold $\tau$: $\mathcal{V}'_{\text{disagreed}} = \{v_i \in \mathcal{V}_{\text{disagreed}} \mid S_{\text{pref}}(v_i) \geq \tau\}$.

> ***Remark***: Directly utilizing LLMs to evaluate "easiness" or "hardness" of unlabeled nodes on TAGs is non-trivial. Our strategy utilize the GNN as a pseudo label judge to effectively identify "easy" and "hard" samples, providing a practical method to identify reliable pseudo labels.

### 3.3 LLM Weakly-Supervised Fine-Tuning on Graphs with GNN Feedback

**Weakly-Supervised Fine-tuning Algorithm.** Based on our selected pseudo-labeled nodes, we propose a weakly-supervised fine-tuning algorithm that fine-tunes LLMs on graphs using a unified objective. Our approach integrates both instruction tuning and preference tuning into a single training framework:

$$\mathcal{L}(\theta) = \mathbb{E}_{(x_i, y_i) \sim \mathcal{D}_{\text{agreed}}}[\mathcal{L}_{\text{IT}}(\theta; x_i, y_i)] + \lambda \mathbb{E}_{(x_i, y_{w,i}, y_{l,i}) \sim \mathcal{D}_{\text{disagreed}'}}[\mathcal{L}_{\text{PT}}(\theta; x_i, y_{w,i}, y_{l,i})] \tag{5}$$

where $\mathcal{D}_{\text{agreed}}$ represents the data distribution over the agreement node set $\mathcal{V}_{\text{agreed}}$, $\mathcal{D}_{\text{disagreed}'}$ represents the data distribution over the selected disagreement node set $\mathcal{V}'_{\text{disagreed}}$, and $\lambda$ controls the contribution of the preference tuning loss. For each selected node $v_i$, we construct the input $x_i$ as the node's textual content formatted with task-specific prompts (detailed prompt templates are provided in Appendix E). For nodes in $\mathcal{V}_{\text{agreed}}$, $y_i$ represents the consensus prediction where both GNN and LLM agree. For nodes in $\mathcal{D}_{\text{disagreed}'}$, $y_{w,i}$ and $y_{l,i}$ represent the GNN's prediction (preferred) and LLM's prediction (dispreferred) respectively.

**LLM Instruction Tuning with LLM-GNN Agreement.** For nodes in the agreement set $\mathcal{V}_{\text{agreed}}$, we apply instruction tuning (Ouyang et al., 2022) to reinforce correct predictions. Given an input node text feature $x_i$ and the agreed pseudo label $y_i$, the instruction tuning loss is defined as:

$$\mathcal{L}_{\text{IT}}(\theta; x_i, y_i) = -\log p_\theta(y_i|x_i), \tag{6}$$

where $p_\theta(y_i|x_i)$ represents the LLM's probability of generating the label $y_i$ given the node text $x_i$, and $\theta$ denotes the model parameters.

**LLM Preference Tuning with LLM-GNN Disagreement.** Directly fine-tuning the LLM on pseudo-labels from the disagreement set via standard instruction tuning is problematic because the disagreement node set contains potentially more label noise compared to the agreement node set. To address this, we reframe the problem on the disagreement set using a preference tuning objective. For each node $v_i \in \mathcal{V}'_{\text{disagreed}}$, we construct a preference pair where the GNN's prediction $y_{w,i}$ serves as the preferred response and the LLM's prediction $y_{l,i}$ serves as the dispreferred response. This formulation enables the model to learn from the relative relationship between competing outputs without requiring absolute correctness from either prediction. The preference tuning objective is formulated as:

$$\mathcal{L}_{\text{PT}}(\theta; x_i, y_{w,i}, y_{l,i}) = -\log \sigma(g_\theta(x_i, y_{w,i}, y_{l,i})) \tag{7}$$

where $x_i$ represents the input prompt for node $v_i$, $y_{w,i}$ and $y_{l,i}$ are the preferred and dispreferred outputs respectively, and $g_\theta(\cdot)$ is a preference function that scores the relative preference between the two outputs. In our implementation, we adopt *Odds Ratio Preference Optimization (ORPO)* (Hong et al., 2024), which uses the log odds ratio as the preference function:

$$g_\theta(x, y_w, y_l) = \log \frac{\text{odds}_\theta(y_w \mid x)}{\text{odds}_\theta(y_l \mid x)} \tag{8}$$

where $\text{odds}_\theta(y \mid x) = P_\theta(y \mid x)/(1 - P_\theta(y \mid x))$. By minimizing this loss over $\mathcal{V}'_{\text{disagreed}}$, the LLM learns to increase the relative likelihood of the GNN's predictions compared to its own initial predictions. This approach mitigates the risk of overfitting to noisy pseudo-labels while still leveraging the valuable disagreement signal to improve model performance.

> ***Remark***: Our framework can be considered as an LLM preference alignment framework by replacing human feedback with signals derived from GNNs. While we implement preference tuning using ORPO (Hong et al., 2024), our approach is also compatible with other methods like such as DPO (Rafailov et al., 2023), SimPO (Meng et al., 2024), and other variants.

## 4 EXPERIMENTS

### 4.1 EXPERIMENTAL SETUP

**Datasets.** We train and evaluate our framework on four widely-used benchmark datasets for few-shot semi-supervised node classification: `Cora` (Yang et al., 2016), `Citeseer` (Sen et al., 2008), `Pubmed` (Sen et al., 2008), `ogbn-arxiv` and `ogbn-products` (Namata et al., 2012). For `Cora`, `Citeseer` and `Pubmed`, which are three most widely used citation networks, we follow the experimental setup of previous work Gasteiger et al. (2018) and split each dataset into training (i.e., $K$ nodes per class for $K$-shot task), validation set, and test set. To evaluate performance on large-scale graphs, we also include `ogbn-arxiv` and `ogbn-products` from the Open Graph Benchmark (OGB) (Hu et al., 2020). Detailed statistics of all datasets are summarized in Appendix A.

**Baselines and Implementation Details.** To evaluate the effectiveness of our proposed framework, we compare it against baseline methods from three primary categories: (1) *Classical GNN Models*:

We include established graph neural network models such as GCN (Kipf & Welling, 2017) and SGC (Wu et al., 2019); (2) *LLM-as-Predictors*: We also include baselines that utilize general-purpose LLMs with various prompting strategies such as zero-shot, chain-of-thought, and neighbor-augmented prompting (Wei et al., 2022; Chen et al., 2024b); (3) *LLM-Graph Methods*: Aligning with our focus, we benchmark against various state-of-the-art LLM-Graph methods, including GLEM (Zhao et al., 2023), TAPE (He et al., 2024), LLM-GNN (Chen et al., 2024c), LLaGA (Chen et al., 2024a) and GraphGPT (Tang et al., 2024). We use GCN (Kipf & Welling, 2017) and Llama-3-8B-Instruct (Grattafiori et al., 2024) as backbone models of our approach. Further descriptions and implementation details on all methods are provided in the Appendix B and C.

## 4.2 FEW-SHOT SEMI-SUPERVISED NODE CLASSIFICATION

Table 1: Node classification accuracy (%) across different datasets and shot settings. Results show mean ± standard deviation performance. **Best** results are bolded, while second-best results are underlined.

| Shot | Method | Cora | Citeseer | Pubmed | ogbn-arxiv | ogbn-products |
|---|---|---|---|---|---|---|
| 3-shot | GCN | 69.45±2.34 | 63.12±1.02 | 65.23±1.67 | 38.33±2.41 | 59.19±0.79 |
| | SGC | 67.21±1.25 | 63.07±0.95 | 65.34±1.18 | 39.16±1.49 | 57.42±1.12 |
| | Zero-Shot | 65.54±0.26 | 58.17±0.64 | 74.51±0.39 | 50.18±1.44 | 75.48±1.54 |
| | Graph-CoT | 63.02±0.77 | 47.23±1.06 | 86.22±2.47 | 49.67±1.23 | 74.15±1.83 |
| | w. Neighbor | 68.72±1.56 | 54.93±1.28 | 74.98±3.16 | 49.28±2.09 | 76.88±1.07 |
| | GLEM | 67.81±0.92 | 53.09±1.39 | 63.85±1.02 | 36.37±1.96 | 52.46±1.93 |
| | TAPE | 73.71±1.86 | 64.96±0.36 | 71.33±0.87 | 48.25±0.77 | 69.64±1.15 |
| | LLM-GNN | 73.85±0.74 | 61.67±0.58 | 66.01±0.63 | 42.36±1.72 | 55.46±0.37 |
| | LLaGA | 54.79±0.91 | 32.93±1.24 | 43.96±1.74 | 29.73±2.82 | 30.67±1.54 |
| | GraphGPT | 57.77±1.62 | 52.34±1.06 | 57.51±1.89 | 31.26±2.73 | 40.83±2.97 |
| | **GNN-as-Judge** | **77.89±1.28** | **73.59±0.64** | **87.12±0.89** | **62.21±1.45** | **81.02±1.23** |
| 5-shot | GCN | 73.69±1.03 | 63.24±0.87 | 70.58±0.49 | 45.67±0.41 | 68.26±1.54 |
| | SGC | 72.48±0.35 | 62.08±0.77 | 70.74±0.94 | 49.89±1.23 | 66.78±1.56 |
| | Zero-Shot | 65.54±0.26 | 58.17±0.64 | 74.51±0.39 | 50.18±1.44 | 75.48±1.54 |
| | Graph-CoT | 63.02±0.77 | 47.23±1.06 | 86.22±2.47 | 49.67±1.23 | 74.15±1.83 |
| | w. Neighbor | 68.72±1.56 | 54.93±1.28 | 74.98±3.16 | 49.28±2.09 | 76.88±1.07 |
| | GLEM | 74.69±0.46 | 61.71±0.58 | 73.29±1.44 | 39.19±1.57 | 56.87±1.70 |
| | TAPE | 74.28±0.81 | 67.73±0.48 | 75.02±0.83 | 55.22±0.48 | 77.44±1.32 |
| | LLM-GNN | 75.61±1.04 | 62.37±1.35 | 74.33±0.95 | 45.74±1.66 | 64.01±0.39 |
| | LLaGA | 62.88±2.19 | 43.71±4.36 | 58.63±1.05 | 33.74±2.45 | 37.29±3.12 |
| | GraphGPT | 60.17±1.44 | 51.83±2.24 | 57.39±3.67 | 36.25±1.87 | 44.78±2.34 |
| | **GNN-as-Judge** | **79.54±0.39** | **74.39±1.63** | **87.49±1.23** | **66.76±0.83** | **81.93±2.21** |
| 10-shot | GCN | 78.22±0.89 | 68.38±1.49 | 75.33±0.94 | 50.95±1.77 | 69.65±0.89 |
| | SGC | 78.49±0.37 | 67.44±0.60 | 74.98±1.91 | 51.89±1.23 | 67.91±0.48 |
| | Zero-Shot | 65.54±0.26 | 58.17±0.64 | 74.51±0.39 | 50.18±1.44 | 75.48±1.54 |
| | Graph-CoT | 63.02±0.77 | 47.23±1.06 | 86.22±2.47 | 49.67±1.23 | 74.15±1.83 |
| | w. Neighbor | 68.72±1.56 | 54.93±1.28 | 74.98±3.16 | 49.28±2.09 | 76.88±1.07 |
| | GLEM | 78.11±0.73 | 66.83±0.61 | 74.17±2.39 | 47.73±1.09 | 60.22±1.89 |
| | TAPE | 79.33±0.57 | 69.39±0.65 | 77.18±1.06 | 60.37±0.92 | 79.53±0.63 |
| | LLM-GNN | 79.39±1.26 | 66.28±0.94 | 76.82±0.57 | 52.74±0.48 | 66.98±0.39 |
| | LLaGA | 69.25±0.97 | 51.22±1.43 | 67.29±2.26 | 45.35±1.74 | 40.55±1.63 |
| | GraphGPT | 61.58±0.77 | 55.40±3.16 | 71.33±2.81 | 48.67±1.89 | 51.46±1.05 |
| | **GNN-as-Judge** | **80.71±0.83** | **74.62±1.35** | **90.17±1.69** | **67.88±1.03** | **82.48±1.56** |

Table 1 presents the comparative performance of our approach against various baseline methods across multiple datasets and experimental settings. Our results demonstrate three key observations. Observation 1: Our proposed *GNN-as-Judge* consistently outperforms all baseline methods across all datasets and experimental settings, demonstrating superior robustness. Observation 2: Our method shows particularly strong performance in low-resource settings, highlighting its effectiveness in real-world applications. In extreme scenarios such as 3-shot and 5-shot settings, *GNN-as-Judge* maintains substantial performance advantages, making it highly practical for domains where labeled data is scarce or expensive to obtain. Observation 3: In low-resource settings, GNN-based methods consistently show strong performance compared to LLM-based approaches. Traditional GNN architectures can effectively exploit graph structure and connectivity patterns even with limited labeled data, while LLM-based methods struggle due to insufficient textual context and unreliable pseudo label generation in these constrained scenarios.

### 4.3 CROSS-DATASET ZERO-SHOT NODE CLASSIFICATION

In this section, we evaluate the zero-shot generalization capabilities of *GNN-as-Judge*. We compared with various *LLMs-as-Predictors* graph learning models on the `ogbn-arxiv` dataset and evaluated their zero-shot performance on `Cora`, `Citeseer`, and `Pubmed` without additional fine-tuning.

Unlike traditional GNNs that require task-specific classification heads, *LLMs-as-Predictors* methods can perform zero-shot learning across different label sets. As shown in Table 2, *GNN-as-Judge* demonstrates superior zero-shot transfer performance across all datasets. It substantially outperforms GraphGPT and LLaGA, which struggle with cross-dataset generalization. Their approach of encoding graph structure into tokens appears to constrain the LLM's generalization capabilities, resulting in performance worse than untuned base LLMs. These results indicate that *GNN-as-Judge* is more robust to distribution shifts between graph datasets and better preserves the LLM's inherent generalization capabilities while incorporating graph-based insights. This makes our approach particularly valuable for practical applications where labeled data may be scarce or available for domains, necessitating models that can effectively generalize to new, unseen graph structures.

Table 2: Zero-Shot Cross-Dataset Node Classification.

| Train → Test | Model | Accuracy |
|---|---|---|
| ogbn-arxiv | LLAGA | 16.24±0.95 |
| ↓ | GraphGPT | 6.29±0.73 |
| **Cora** | **GNN-as-Judge** | **68.27±0.91** |
| ogbn-arxiv | LLAGA | 14.72±1.12 |
| ↓ | GraphGPT | 5.37±0.84 |
| **Citeseer** | **GNN-as-Judge** | **56.67±0.89** |
| ogbn-arxiv | LLAGA | 30.52±1.18 |
| ↓ | GraphGPT | 10.54±1.05 |
| **Pubmed** | **GNN-as-Judge** | **83.41±0.76** |

### 4.4 ANALYSIS ON PSEUDO LABEL SELECTION

To evaluate the effectiveness of our influence-guided node selection strategy, we conduct a comprehensive analysis comparing different pseudo label selection approaches. Figure 2 illustrates the accuracy of 1,500 pseudo labels selected using various methods across three representative datasets under 3-shot setting and then annotated by GCN (Kipf & Welling, 2017).

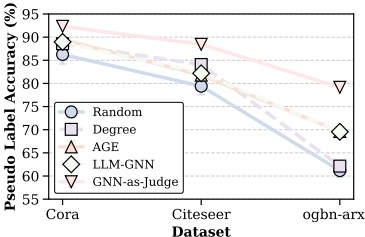

Figure 2: Comparison of pseudo label selection strategies across datasets.

We apply the same *GNN-as-Judge* filtering process after initial node selection. We analyze our *GNN-as-Judge* approach alongside several alternative selection strategies: (1) *Random*: Randomly selecting pseudo labels; (2) *Degree* (Page et al., 1999): Selecting nodes based on their degree centrality in the graph; (3) *AGE* (Cai et al., 2017): Using graph embedding to identify representative nodes; and (4) *LLM-GNN* (Chen et al., 2024c): Specifically, we use the proposed difficulty-aware strategy combined with *AGE* (Cai et al., 2017). As shown in Figure 2, our influence-guided selection consistently outperforms all baseline approaches across datasets, achieving the highest pseudo label accuracy after applying the same filtering process. The results highlight that structural influence from labeled nodes provides a more effective criterion for pseudo label selection compared to simple graph topology measures. More detailed quantitative analysis of pseudo label selection is provided in Appendix F.

### 4.5 ABLATION STUDY

In this section, we conduct an ablation study to analyze the contribution of key modules from our proposed *GNN-as-Judge*.

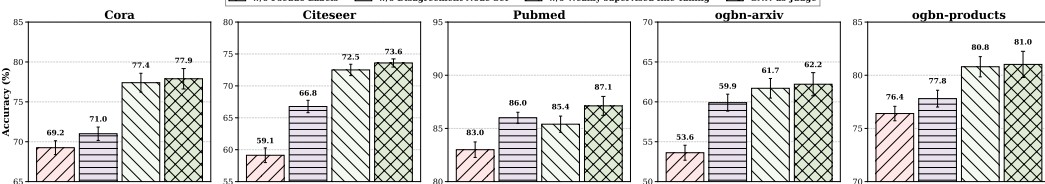

Figure 3: Ablation study demonstrating the contribution of each component in our framework.

- **Effectiveness of Pseudo Label.** We first examine the contribution of pseudo labels by evaluating the *w/o Pseudo Labels* variant that only performs supervised fine-tuning on the labeled set without incorporating pseudo-labeled nodes. As shown in Figure 3, removing pseudo labels leads to

significant performance degradation across all datasets. This demonstrates that our pseudo-labeling strategy effectively expands the training set with high-quality pseudo labels.

- **Effectiveness of Disagreement Node Set.** We then evaluate the importance of the disagreement node set through the *w/o Disagreement Node Set* variant by excluding these challenging examples from our training data. Removing the disagreement node set leads to significant performance degradation. These nodes, identified through prediction disagreements between different model variants, represent the most informative hard examples that can provide additional learning signals.

- **Effectiveness of Weakly-Supervised Fine-Tuning.** Finally, we assess the contribution of our proposed weakly supervised fine-tuning strategy through the *w/o Weakly supervised fine-tuning* variant by replacing it with standard instruction tuning. Our full *GNN-as-Judge* approach consistently outperforms standard instruction tuning on selected nodes with particularly notable gains on `Pubmed` due to potentially more label noise in the disagreement set. Unlike standard instruction tuning on pseudo-labeled data, our approach is better equipped to handle the inherent uncertainty or potential noise present in the pseudo labels of the selected nodes.

## 4.6 SENSITIVITY AND TRAINING TIME ANALYSIS

In this section, we evaluate the framework's sensitivity to different hyper-parameters and analyze the computational efficiency. Additional experiments and analysis are provided in Appendix G.

**Sensitivity Analysis.** We investigate the impact of two critical hyperparameters in *GNN-as-Judge*: the top-$K$ selection for unlabeled nodes in pseudo-labeling and the preference score threshold $\tau$ for disagreement node filtering. This analysis is essential for understanding the robustness and the scalability of our approach.

For the top-$K$ analysis, we vary the number of selected unlabeled nodes within the range $K \in \{500, 1000, 1500, 2000, 2500\}$. Figure 4 demonstrates that *GNN-as-Judge* exhibits scalability properties, with performance generally improving as more unlabeled nodes are incorporated into the pseudo-labeling process and then used for training. However, the performance increase is moderate as more data points are incorporated, suggesting diminishing returns beyond a certain threshold while maintaining computational efficiency.

Figure 4: Sensitivity analysis of hyperparameters: top-$K$ unlabeled nodes (left) and preference score threshold $\tau$ (right) across three benchmark datasets.

For the preference score threshold $\tau$, we examine values in the range $\tau \in \{0.1, 0.3, 0.5, 0.7, 0.9\}$ to understand how preference score filtering affects model performance. Notably, the performance variations are relatively small when $\tau$ ranges from 0.5 to 0.9. This robustness is particularly valuable for practical applications where precise hyperparameter tuning may be challenging.

**Training Time Analysis.** Figure 5 presents a comprehensive comparison of the computational efficiency and accuracy trade-offs across different methods. The scatter plot illustrates the relationship between total training time on a single NVIDIA A100 80GB GPU for the entire pipeline and achieved accuracy for each approach under 3-shot settings for the `ogbn-arxiv` dataset. The usage of LLMs extends the training time, but our method gains substantial accuracy improvements that justify the computational overhead.

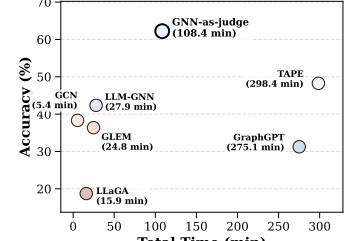

Figure 5: Training time versus accuracy for all methods.

## 5 CONCLUSION

In this paper, we present *GNN-as-Judge*, a novel framework that addresses the challenge of applying LLMs to few-shot semi-supervised graph learning. Our approach leverages complementary strengths from both GNNs and LLMs through three key mechanisms: (i) a subset selection strategy that

identifies nodes with most information from labeled nodes for pseudo labeling; (ii) a strategic pseudo-label selection that identifies both reliable and challenging nodes, and (iii) a weakly-supervised fine-tuning strategy combining instruction tuning with preference tuning. Experiments across multiple datasets demonstrate that *GNN-as-Judge* consistently outperforms both GNN architectures and LLM-based methods, particularly in low-resource settings.

## REPRODUCIBILITY STATEMENT

To ensure reproducibility of our results, we provide our complete source code at `https://github.com/rux001/GNN-as-Judge`. Complete implementation details are provided in Appendix C.

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

APPENDIX

This Appendix provides comprehensive supplementary material for the submitted paper including detailed information across the following sections:

- **Section A:** Dataset statistics and descriptions including comprehensive analysis of evaluation datasets with node/edge counts, feature dimensions, and label space characterization.

- **Section B:** Comprehensive baseline method descriptions covering Classical GNN Models, LLM-as-Predictors approaches, and state-of-the-art LLM-Graph integration methods.

- **Section C:** Implementation details for our approach and all baselines, including hyperparameter settings, training configurations, and computational environment specifications for reproducibility.

- **Section D:** Theoretical analysis with complete proofs for Theorem 1 and Theorem 2.

- **Section E:** Detailed prompt templates used for all LLM-based methods including our *GNN-as-Judge* framework and baseline approaches.

- **Section F:** Quantitative analysis of pseudo-label selection including agreement patterns, accuracy distributions, diversity metrics, and selection strategy comparisons.

- **Section G:** Additional experimental results covering sensitivity analysis, integration with various LLMs, performance on heterophilic graphs.

- **Section H:** Discussion of limitations and potential future research directions for the *GNN-as-Judge* framework.

- **Section I:** Extended related work.

- **Section J:** Disclosure of Large Language Model usage in manuscript preparation for transparency and reproducibility standards.

## A    DATASET

### A.1    DATASET STATISTICS

Table 3: Summary statistics of the evaluation datasets.

| Dataset | Nodes | Edges | Features | Classes |
|---|---|---|---|---|
| Cora | 2,708 | 5,429 | 1,433 | 7 |
| Citeseer | 3,327 | 4,732 | 3,703 | 6 |
| Pubmed | 19,717 | 44,338 | 500 | 3 |
| ogbn-arxiv | 169,343 | 1,166,243 | 128 | 40 |
| ogbn-products(subset) | 54,025 | 74,420 | 100 | 47 |

Table 3 presents the detailed statistics of the datasets used in our experiments, including the number of nodes, edges, features, and classes for each dataset.

### A.2    DATASET DESCRIPTIONS

**Cora** (Yang et al., 2016). The Cora dataset comprises 2,708 scientific publications classified into one of seven machine learning research categories. The citation network consists of 5,429 links, where papers were selected such that every paper citeps or is citepd by at least one other paper in the final corpus.

**citeseer** (Sen et al., 2008). The citeseer dataset contains 3,327 computer science papers classified into 6 categories across different CS research areas. The citation network includes 4,732 links, with each paper represented by 3,703-dimensional features derived from bag-of-words representation of the paper content.

**Pubmed** (Sen et al., 2008). The Pubmed dataset consists of 19,717 scientific publications from the PubMed database pertaining to diabetes research, classified into one of three diabetes-related

categories. The citation network consists of 44,338 links, with each paper represented by 500-dimensional features extracted from the paper abstracts.

**ogbn-arxiv** (Hu et al., 2020). The `ogbn-arxiv` dataset is a directed graph representing the citation network between computer science arXiv papers indexed by Microsoft Academic Graph (MAG). Each node represents an arXiv paper, and each directed edge indicates that one paper citeps another.

**ogbn-products** (Hu et al., 2020). The `ogbn-products` dataset represents an Amazon product co-purchasing network with product descriptions as raw text. Nodes represent products sold on Amazon, and edges between products indicate co-purchasing relationships. We use a subset of 54,025 products for computational efficiency following previous work (He et al., 2024) while maintaining the dataset's key characteristics.

## A.3 LABEL SPACE

Within each dataset, nodes are labeled according to their academic category, as detailed in Table 4. For example, the arXiv dataset includes 40 computer science sub-categories such as cs.AI (Artificial Intelligence) and cs.DB (Databases).

Table 4: Label spaces of the datasets used in our experiments.

| Dataset | Label Space |
|---|---|
| Cora | Rule Learning, Neural Networks, Case Based, Genetic Algorithms, Theory, Reinforcement Learning, Probabilistic Methods |
| citeseer | Agents, ML (Machine Learning), IR (Information Retrieval), DB (Databases), HCI (Human-Computer Interaction), AI (Artificial Intelligence) |
| Pubmed | Experimentally induced diabetes, Type 1 diabetes, Type 2 diabetes |
| ogbn-arxiv | cs.NA, cs.MM, cs.LO, cs.CY, cs.CR, cs.DC, cs.HC, cs.CE, cs.NI, cs.CC, cs.AI, cs.MA, cs.GL, cs.NE, cs.SC, cs.AR, cs.CV, cs.GR, cs.ET, cs.SY, cs.CG, cs.OH, cs.PL, cs.SE, cs.LG, cs.SD, cs.SI, cs.RO, cs.IT, cs.PF, cs.CL, cs.IR, cs.MS, cs.FL, cs.DS, cs.OS, cs.GT, cs.DB, cs.DL, cs.DM |
| products | Home & Kitchen, Health & Personal Care, Beauty, Sports & Outdoors, Books, Patio Lawn & Garden, Toys & Games, CDs & Vinyl, Cell Phones & Accessories, Grocery & Gourmet Food, Arts Crafts & Sewing, Clothing Shoes & Jewelry, Electronics, Movies & TV, Software, Video Games, Automotive, Pet Supplies, Office Products, Industrial & Scientific, Musical Instruments, Tools & Home Improvement, Magazine Subscriptions, Baby Products, Appliances, Kitchen & Dining, Collectibles & Fine Art, All Beauty, Luxury Beauty, Amazon Fashion, Computers, All Electronics, Purchase Circles, MP3 Players & Accessories, Gift Cards, Office & School Supplies, Home Improvement, Camera & Photo, GPS & Navigation, Digital Music, Car Electronics, Baby, Kindle Store, Kindle Apps, Furniture & Decor |

## B BASELINE METHODS

To evaluate the effectiveness of our proposed framework, we compare it against established methods from three primary categories: (1) *Classical GNN Models*, (2) *LLM-as-Predictors*, and (3) *LLM-Graph Methods*. These baselines represent diverse approaches to handling Text-Attributed Graphs, ranging from traditional graph neural networks to modern large language model-based methods.

### B.1 CLASSICAL GNN MODELS

We include established graph neural network architectures that rely primarily on graph structure and node features without leveraging the reasoning capabilities of language models:

- **GCN** (Kipf & Welling, 2017): Graph Convolutional Networks perform neighborhood aggregation through spectral convolutions, using a simple and effective message-passing scheme.
- **SGC** (Wu et al., 2019): Simple Graph Convolution simplifies GCNs by removing nonlinearities between layers and collapsing weight matrices, resulting in a single linear transformation followed by a softmax classifier, while maintaining competitive performance with significantly reduced computational complexity.

## B.2 LLM-as-Predictors Methods

We include baselines that utilize general-purpose LLMs with various prompting strategies for direct downstream classification tasks. In this paradigm, a node's textual and structural information, along with task-specific instructions, are tokenized and input into an LLM for prediction:

- **Zero-shot Prompting**: Direct application of LLMs without task-specific training, relying on the model's pre-trained knowledge.
- **Graph Chain-of-Thought** (Wei et al., 2022; Chen et al., 2024b): Enables step-by-step reasoning by breaking down complex graph-related problems into sequential reasoning steps.
- **Neighbor-Augmented Prompting** (Chen et al., 2024b): Enriches prompts with structural information from node neighborhoods, providing context about graph topology to enhance zero-shot performance.

## B.3 LLM-Graph Methods

Aligning with our focus, we benchmark against various state-of-the-art methods that effectively integrate LLMs with graph neural networks for enhanced performance on Text-Attributed Graphs:

- **GLEM** (Zhao et al., 2023): Combines language models with graph neural networks through a unified framework that leverages both textual and structural information.
- **TAPE** (He et al., 2024): A text-attributed graph learning framework that effectively harnesses the power of language models for graph understanding.
- **LLM-GNN** (Chen et al., 2024c): Integrates large language models with graph neural networks for improved node classification performance.
- **LLaGA** (Chen et al., 2024a): Employs a multi-step approach where node text is first encoded via a language model, then processed through a GNN, concatenated across layers, projected into the LLM's dimensionality, and finally combined with instructions for label prediction. Only the projection layer parameters are tuned using next-token-prediction loss.
- **GraphGPT** (Tang et al., 2024): Implements a comprehensive framework with three distinct pre-training and instruction tuning stages to effectively integrate graph structure with language understanding.

## C Implementation Details

This section provides comprehensive details about the implementation, hyperparameter settings, and training procedures used in our experiments.

### C.1 Implementation Details for GNN-as-Judge

**Data Splits.** For the `Cora`, `citeseer`, and `Pubmed` datasets, we follow standard node classification protocols using 3-shot, 5-shot, and 10-shot settings, where *n*-shot refers to *n* labeled nodes per class for training. We use 500 nodes for validation and randomly select 1,000 nodes for testing from the remaining nodes. For `ogbn-arxiv` and `ogbn-products` dataset, we adopt the original split for both validation and testing provided by (Hu et al., 2020). For the training split, we randomly select 3-shot, 5-shot, and 10-shot settings by sampling *n* nodes from each class within the original training set to create our labeled node sets.

**GNN Component.** For the GNN component of our *GNN-as-Judge* framework and other GNN-based baselines, we implement a 2-layer GCN architecture (Kipf & Welling, 2017) with 64-dimensional

hidden representations. We perform a limited grid search on dropout rates, exploring values in [0.3, 0.5, 0.7], and include batch normalization in our architecture. We set the learning rate to 1e-2 and train for up to 200 epochs with a patience of 100 for early stopping. For optimization, we use the Adam optimizer with a weight decay of 5e-4, following standard practices in graph neural network training.

**LLM Component.** Our *GNN-as-Judge* framework utilizes Llama-3-8B-Instruct (Grattafiori et al., 2024) as the base model. We implement parameter-efficient fine-tuning using LoRA (Hu et al., 2022) with rank 8 and alpha value of 16, which enables efficient adaptation of the pre-trained weights while maintaining computational feasibility. We set the dropout rate to 0.1 and use a batch size of 8.

- **Instruction Tuning Configuration.** We use a learning rate of $5 \times 10^{-6}$ for the Llama-3 model and train for 10 epochs during the instruction tuning stage.

- **Weakly-Supervised Fine-Tuning Configuration.** We maintain a learning rate of $1 \times 10^{-5}$ with 8 epochs for all datasets. The hyperparameter $\lambda$ in Eq. 5, which balances instruction tuning and preference tuning losses, is set to 0.1 across all datasets and settings based on our parameter sensitivity analysis.

**Selection Hyperparameters.** For our proposed *GNN-as-Judge* pseudo-label selection mechanism described in Section 3, we employ several key hyperparameters. We set the top-K selection parameter to 1,500 influential nodes for all datasets to balance computational efficiency with pseudo-labeling coverage. For large-scale datasets (`ogbn-arxiv`, `ogbn-products`), we extract subgraphs around labeled nodes. The preference score threshold $\tau$ is set to 0.7 across all datasets.

**Computational Environment.** All experiments are conducted on NVIDIA A100 GPUs with 80GB memory.

## C.2 Implementation Details for Other Baselines

- **GLEM** (Zhao et al., 2023): Following previous work (Wu et al., 2025), we set the number of EM iterations to 1 and the pseudo-labeling ratio to 0.5. For the GNN module within GLEM, we use 2 layers with 64-dimensional hidden representations. For the LM module, we use RoBERTa (Liu et al., 2019) as the base model with LoRA optimization and a batch size of 32. The LM is pre-trained first across all datasets before joint training with the GNN component.

- **TAPE** (He et al., 2024): We utilize the provided prompt templates from the original paper to generate explanations using Llama-3-8B-Instruct (Grattafiori et al., 2024). We use RoBERTa (Liu et al., 2019) as the language model component, which is fine-tuned using LoRA with default parameter settings, while the GNN configuration remains consistent with our method.

- **LLM-GNN** (Chen et al., 2024c): To ensure fair comparison with this baseline, we adapt the original zero-shot approach to our few-shot setting. We use instruction-tuned Llama-3-8B-Instruct (Grattafiori et al., 2024) on labeled data as the annotator. We employ the DA-AGE method proposed in the original paper for pseudo-label selection and train the GNN using both labeled data and pseudo-labeled data together.

- **LLaGA** (Chen et al., 2024a): We use HO templates for all experiments with the number of hops set to 4. We use RoBERTa (Liu et al., 2019) as the text encoder. The linear projection layer $\phi_\theta(\cdot)$ consists of a 2-layer MLP with a hidden dimension of 2048. The batch size is set to 64 and learning rate to $1 \times 10^{-4}$. The number of training epochs is set to 10 for all settings.

- **GraphGPT** (Tang et al., 2024): Following previous work (Wu et al., 2025), we exclude the text-graph grounding stage since the inclusion of this stage does not consistently lead to performance improvements. For the self-supervised instruction tuning stage, we construct self-supervised training data for each dataset to perform dataset-specific graph matching tasks. In the task-specific instruction tuning stage, we utilize the training data to create ⟨instruction, ground-truth label⟩ pairs following the original prompt design. The training parameters for self-supervised instruction tuning stage include 2 epochs with a learning rate of $1 \times 10^{-4}$ and a batch size of 16. For task-specific instruction tuning stage, we train for 10 epochs with a learning rate of $1 \times 10^{-4}$ and a batch size of 32.

# D  THEORETICAL ANALYSIS

## D.1  PROOF OF THEOREM 1

Following previous study (Xu et al., 2018; Huang & Zitnik, 2020; Wang & Leskovec, 2020), we use GCN (Kipf & Welling, 2017) as as the exemplar GNN model. The propagation mechanism for the $l$-th layer in GCN is formulated as: $\mathbf{H}^{(l+1)} = \sigma(\hat{\mathbf{A}}\mathbf{H}^{(l)}\mathbf{W}^{(l)})$, where $\mathbf{H}^{(l)}$ represents the node feature matrix and $\mathbf{W}^{(l)}$ denotes the learnable parameter matrix at layer $l$. The term $\hat{\mathbf{A}} = \mathbf{D}^{-1}\mathbf{A}$ corresponds to the row-normalized adjacency matrix. To simplify our theoretical derivations, we adopt the standard assumptions from (Wang & Leskovec, 2020), that the activation function $\sigma$ acts as the identity operator and the weight matrix $W$ is the identity matrix.

**Theorem 1.** *Let $\mathcal{P}_{v_i,v_j}$ denote the set of all paths between nodes $v_i$ and $v_j$, and let $\mathcal{P}^*_{v_i,v_j} \subseteq \mathcal{P}_{v_i,v_j}$ denote the set of shortest paths. For any path $t \in \mathcal{P}_{v_i,v_j}$, let $D^t_{GM}$ be the geometric mean of node degrees occurring on path $t$ defined as $D^t_{GM} = \left(\prod_{v_k \in t} D_{v_k v_k}\right)^{1/|t|}$, where $|t|$ is the path length and $D_{v_k v_k}$ is the degree of node $v_k$. Define $D^*_{GM} = \min_{t \in \mathcal{P}^*_{v_i,v_j}}\{D^t_{GM}\}$ as the minimum geometric mean among shortest paths, $h^* = d(v_i, v_j)$ as the shortest path distance between $v_i$ and $v_j$, and $|\mathcal{P}^*_{v_i,v_j}|$ as the number of shortest paths between $v_i$ and $v_j$. Then, the node influence $I_{v_i,v_j}$ from $v_i$ to $v_j$ satisfies:*

$$I_{v_i,v_j} = \left\|\partial\mathbf{x}^{(\infty)}_{v_j}/\partial\mathbf{x}^{(\infty)}_{v_i}\right\| \leq \frac{|\mathcal{P}^*_{v_i,v_j}|}{(D^*_{GM})^{h^*}} \tag{9}$$

*Proof.* According to the GCN propagation rule, the final representation of node $v_j$ is:

$$\mathbf{x}^{(\infty)}_{v_j} = \frac{1}{D_{v_j v_j}} \sum_{v_k \in \mathcal{N}(v_j)} a_{v_j v_k} \mathbf{x}^{(\infty)}_{v_k} \tag{1}$$

where $\mathcal{N}(v_j)$ denotes the neighboring nodes of $v_j$, $a_{v_j v_k}$ is the edge weight (typically 1 for unweighted graphs), and $D_{v_j v_j}$ is the degree of node $v_j$.

By recursive substitution incorporating neighbors at multiple hops:

$$\mathbf{x}^{(\infty)}_{v_j} = \frac{1}{D_{v_j v_j}} \sum_{v_k \in \mathcal{N}(v_j)} a_{v_j v_k} \frac{1}{D_{v_k v_k}} \sum_{v_l \in \mathcal{N}(v_k)} a_{v_k v_l} \cdots \frac{1}{D_{v_m v_m}} \sum_{v_o \in \mathcal{N}(v_m)} a_{v_m v_o} \mathbf{x}^{(\infty)}_{v_o} \tag{2}$$

The node influence $I_{v_i,v_j} = \left\|\frac{\partial\mathbf{x}^{(\infty)}_{v_j}}{\partial\mathbf{x}^{(\infty)}_{v_i}}\right\|$ can be computed by taking the derivative with respect to $\mathbf{x}^{(\infty)}_{v_i}$. Since we are calculating the gradient between two feature vectors $\mathbf{x}^{(\infty)}_{v_j}$ and $\mathbf{x}^{(\infty)}_{v_i}$, the partial derivative on nodes that are not in the paths $p_1, ..., p_m$ between node $v_i$ and $v_j$ becomes 0 and these nodes are removed:

$$\frac{\partial\mathbf{x}^{(\infty)}_{v_j}}{\partial\mathbf{x}^{(\infty)}_{v_i}} = \sum_{t \in \mathcal{P}_{v_j,v_i}} \left(\frac{1}{D_{v_j v_j}} a_{v_j, p^t_1} \frac{1}{D_{p^t_1, p^t_1}} a_{p^t_1, p^t_2} \cdots \frac{1}{D_{p^t_{n_t}, p^t_{n_t}}} a_{p^t_{n_t}, v_i}\right) \frac{\partial\mathbf{x}^{(\infty)}_{v_i}}{\partial\mathbf{x}^{(\infty)}_{v_i}} \tag{3}$$

We separate the scalar terms and the derivative term within the matrix norm and use the absolute homogeneous property ($\|\alpha A\| = |\alpha|\|A\|$) of the matrix norm:

$$\left|\frac{\partial\mathbf{x}^{(\infty)}_{v_j}}{\partial\mathbf{x}^{(\infty)}_{v_i}}\right| = \left|\sum_{t \in \mathcal{P}_{v_j,v_i}} \left(\frac{1}{D_{v_j v_j}} \frac{1}{D_{p^t_1, p^t_1}} \cdots \frac{1}{D_{p^t_{n_t}, p^t_{n_t}}} a_{v_j, p^t_1} a_{p^t_1, p^t_2} \cdots a_{p^t_{n_t}, v_i}\right)\right| \cdot \left|\frac{\partial\mathbf{x}^{(\infty)}_{v_i}}{\partial\mathbf{x}^{(\infty)}_{v_i}}\right| \tag{4}$$

Since the Jacobian of the same vectors $\frac{\partial\mathbf{x}^{(\infty)}_{v_i}}{\partial\mathbf{x}^{(\infty)}_{v_i}}$ is the identity matrix $I$ and for any subordinate norm ($\|A\| = \sup_{\|x\|=1}\{\|Ax\|\}$), we have $\|I\| = 1$:

$$I_{v_i,v_j} = \left| \sum_{t \in \mathcal{P}_{v_j,v_i}} \frac{1}{D_{v_j v_j} D_{p_1^t,p_1^t} \cdots D_{p_{n_t}^t,p_{n_t}^t}} a_{v_j,p_1^t} a_{p_1^t,p_2^t} \cdots a_{p_{n_t}^t,v_i} \right| \tag{5}$$

Using the triangle inequality and identifying the maximum term among all paths:

$$I_{v_i,v_j} \leq \sum_{t \in \mathcal{P}_{v_j,v_i}} \left| \frac{1}{D_{v_j v_j} D_{p_1^t,p_1^t} \cdots D_{p_{n_t}^t,p_{n_t}^t}} a_{v_j,p_1^t} a_{p_1^t,p_2^t} \cdots a_{p_{n_t}^t,v_i} \right| \tag{6}$$

$$\leq |\mathcal{P}_{v_j,v_i}| \cdot \max_{t \in \mathcal{P}_{v_j,v_i}} \left| \frac{1}{D_{v_j v_j} D_{p_1^t,p_1^t} \cdots D_{p_{n_t}^t,p_{n_t}^t}} a_{v_j,p_1^t} a_{p_1^t,p_2^t} \cdots a_{p_{n_t}^t,v_i} \right| \tag{7}$$

For binary networks where edge weights $a = 1$ (non-negative), and focusing on the path $p^{t^*}$ that maximizes the influence contribution:

$$I_{v_i,v_j} \leq |\mathcal{P}_{v_j,v_i}| \cdot \frac{1}{D_{v_j v_j} D_{p_1^{t^*},p_1^{t^*}} \cdots D_{p_{n^*}^{t^*},p_{n^*}^{t^*}}} \tag{8}$$

Expressing the degree terms in geometric mean format where $|t^*|$ is the length of path $t^*$:

$$I_{v_i,v_j} \leq |\mathcal{P}_{v_j,v_i}| \cdot \frac{1}{(D_{GM}^{t^*})^{|t^*|}} \tag{9}$$

$\square$

## D.2    PROOF OF THEOREM 2

**Theorem 2.** *Consider $\mathcal{V}_{selected}$ with ground truth labels $\mathbf{y}^*$. Let $p_{LLM}$ and $p_{GNN}$ denote the individual accuracies of the LLM and GNN respectively. Under conditional independence of model errors due to different inductive biases and uniform error distribution across incorrect classes, the accuracy of the agreement set satisfies:*

$$P(y_i^*|v_i \in \mathcal{V}_{agreed}) \geq \frac{p_{LLM} \cdot p_{GNN}}{p_{LLM} \cdot p_{GNN} + \frac{(1-p_{LLM})(1-p_{GNN})}{C-1}} \tag{10}$$

*where $C$ is the number of classes. Furthermore, this lower bound exceeds $\max(p_{LLM}, p_{GNN})$ when both models perform better than random guessing.*

*Proof.* We establish a lower bound on the accuracy of the agreement set by analyzing the probability of correct predictions under model agreement.

The probability that both models agree on their predictions decomposes as:

$$P(\text{agreement}) = P(y_i^{\text{LLM}} = y_i^{\text{GNN}} = y_i^*) + P(y_i^{\text{LLM}} = y_i^{\text{GNN}} \neq y_i^*) = p_{\text{LLM}} \cdot p_{\text{GNN}} + \epsilon \tag{11}$$

where $\epsilon$ represents the probability of agreeing on an incorrect prediction.

Under conditional independence and uniform error distribution across $C - 1$ incorrect classes:

$$\epsilon = \sum_{j \neq y_i^*} P(y_i^{\text{LLM}} = j | y_i^{\text{LLM}} \neq y_i^*) \cdot P(y_i^{\text{GNN}} = j | y_i^{\text{GNN}} \neq y_i^*) \times P(y_i^{\text{LLM}} \neq y_i^*) \cdot P(y_i^{\text{GNN}} \neq y_i^*) \tag{12}$$

$$= \sum_{j \neq y_i^*} \frac{1}{C-1} \cdot \frac{1}{C-1} \cdot (1 - p_{\text{LLM}}) \cdot (1 - p_{\text{GNN}}) = \frac{(1 - p_{\text{LLM}})(1 - p_{\text{GNN}})}{C-1} \tag{13}$$

By Bayes' theorem, the accuracy of the agreement set is:

$$P(y_i^*|\text{agreement}) = \frac{P(\text{agreement}|y_i^*) \cdot P(y_i^*)}{\P(\text{agreement})} = \frac{p_{\text{LLM}} \cdot p_{\text{GNN}}}{p_{\text{LLM}} \cdot p_{\text{GNN}} + \epsilon} \tag{14}$$

$$= \frac{p_{\text{LLM}} \cdot p_{\text{GNN}}}{p_{\text{LLM}} \cdot p_{\text{GNN}} + \frac{(1-p_{\text{LLM}})(1-p_{\text{GNN}})}{C-1}} \tag{15}$$

This establishes the lower bound stated in the theorem.

To show this lower bound exceeds individual model performance, we prove it for $p_{\text{LLM}}$:

$$\frac{p_{\text{LLM}} \cdot p_{\text{GNN}}}{p_{\text{LLM}} \cdot p_{\text{GNN}} + \frac{(1-p_{\text{LLM}})(1-p_{\text{GNN}})}{C-1}} > p_{\text{LLM}} \tag{16}$$

Cross-multiplying and simplifying:

$$p_{\text{LLM}} \cdot p_{\text{GNN}} > p_{\text{LLM}} \left( p_{\text{LLM}} \cdot p_{\text{GNN}} + \frac{(1-p_{\text{LLM}})(1-p_{\text{GNN}})}{C-1} \right) \tag{17}$$

$$0 > p_{\text{LLM}}^2 \cdot p_{\text{GNN}} - p_{\text{LLM}} \cdot p_{\text{GNN}} + \frac{p_{\text{LLM}}(1-p_{\text{LLM}})(1-p_{\text{GNN}})}{C-1} \tag{18}$$

$$p_{\text{LLM}} \cdot p_{\text{GNN}}(1-p_{\text{LLM}}) > \frac{p_{\text{LLM}}(1-p_{\text{LLM}})(1-p_{\text{GNN}})}{C-1} \tag{19}$$

Dividing by $p_{\text{LLM}}(1-p_{\text{LLM}}) > 0$:

$$p_{\text{GNN}} > \frac{1-p_{\text{GNN}}}{C-1} \Rightarrow p_{\text{GNN}}(C-1) > 1 - p_{\text{GNN}} \Rightarrow p_{\text{GNN}} \cdot C > 1 \Rightarrow p_{\text{GNN}} > \frac{1}{C} \tag{20}$$

By symmetry, the same holds for $p_{\text{LLM}} > \frac{1}{C}$. Therefore, when both models perform better than random guessing, the lower bound exceeds both individual accuracies:

$$P(y_i^*|v_i \in \mathcal{V}_{\text{agreed}}) \geq \frac{p_{\text{LLM}} \cdot p_{\text{GNN}}}{p_{\text{LLM}} \cdot p_{\text{GNN}} + \frac{(1-p_{\text{LLM}})(1-p_{\text{GNN}})}{C-1}} > \max(p_{\text{LLM}}, p_{\text{GNN}}) \tag{21}$$

$\square$

# E PROMPTS

This section provides detailed prompt templates used for all LLM-based methods in our experiments. We use ⟨raw_text⟩ to denote the node's original textual content, ⟨labels⟩ to represent the dataset-specific label space, and ⟨graph⟩ for tokenized graph context when applicable.

## E.1 PROMPTS FOR GNN-AS-JUDGE

For our proposed method, we use straightforward instruction-following prompts tailored to each dataset:

---

**GNN-as-Judge Prompt Templates**

**Cora:** Given a node-centered graph with centric node description: ⟨raw_text⟩, each node represents a paper, we need to classify the center node into 7 classes: Case_Based, Genetic_Algorithms, Neural_Networks, Probabilistic_Methods, Reinforcement_Learning, Rule_Learning, Theory, please tell me which class the center node belongs to?

**citeseer:** Given a node-centered graph with centric node description: ⟨raw_text⟩, each node represents a paper, we need to classify the center node into 6 classes: Agents, ML (Machine Learning), IR (Information Retrieval), DB (Databases), HCI (Human-Computer Interaction), AI (Artificial Intelligence), please tell me which class the center node belongs to?

**Pubmed:** Given a node-centered graph with centric node description: ⟨raw_text⟩, each node represents a paper about Diabetes, we need to classify the center node into 3 classes: Experimentally induced diabetes, Type 1 diabetes, Type 2 diabetes, please tell me which class the center node belongs to?

**ogbn-arxiv:** Given a node-centered graph with centric node description: ⟨raw_text⟩, we need to classify the center node into 40 arXiv CS sub-categories: cs.AI(Artificial Intelligence), cs.CV(Computer Vision and Pattern Recognition), cs.LG(Machine Learning), cs.CL(Computation and Language), cs.NE(Neural and Evolutionary Computing), ..., please tell me which class the center node belongs to?

**ogbn-products:** Given a node-centered graph with centric node description: ⟨raw_text⟩, each node represents a product, we need to classify the center node into 47 classes: Home & Kitchen, Health & Personal Care, Beauty, Sports & Outdoors, Books, Electronics, ..., please tell me which class the center node belongs to?

---

### E.2 PROMPTS FOR GRAPHGPT

GraphGPT uses graph-aware templates with structured graph tokens:

---

**GraphGPT Prompt Templates**

**Citation Networks (Cora, citeseer, Pubmed, ogbn-arxiv):**
Given a citation graph: ⟨graph⟩ where the 0-th node is the target paper, with the following information: ⟨raw_text⟩. Question: Which of the following specific research does this paper belong to: ⟨labels⟩. Directly give the full name of the most likely category of this paper.

**E-commerce Network (ogbn-products):**
Given an e-commerce network: ⟨graph⟩ where the 0-th node is the target product, with the following information: ⟨raw_text⟩. Question: We need to classify the center product into 47 classes: ⟨labels⟩. Directly tell me which product category the center product belongs to.

---

### E.3 PROMPTS FOR LLAGA

LLaGA employs HO templates with multi-hop structural information:

---

**LLaGA Prompt Templates**

**Citation Networks (Cora, citeseer, Pubmed, ogbn-arxiv):**
Given a node-centered graph: ⟨graph⟩, each node represents a paper, we need to classify the center node into [N] classes: ⟨labels⟩, please tell me which class the center node belongs to?

**E-commerce Network (ogbn-products):**
Given a node-centered graph: ⟨graph⟩, each node represents a product, we need to classify the center node into 47 classes: Home & Kitchen, Health & Personal Care, Beauty, Sports & Outdoors, Books, Patio Lawn & Garden, Electronics, ..., please tell me which class the center node belongs to?

---

### E.4 PROMPTS FOR ZERO-SHOT LLM METHODS

For direct LLM inference without fine-tuning, we employ several prompting strategies:

---

**Zero-Shot Prompting Templates**

**Chain-of-Thought:**
Question: Which of the following types does this [paper/product] belong to? Here are the [N] categories: ⟨labels⟩. Let's think about it step by step. Analyze the content of the node and choose one appropriate category. Output format: ⟨reason:⟩, ⟨classification:⟩

**Neighbor-Augmented:**
Given the information of the node: ⟨raw_text⟩. Given the information of its neighbors ⟨raw_text⟩. Here I give you the content of the node itself and the information of its 2-hop neighbors. The relation between the node and its neighbors is ["citation"/"co-purchase"]. Question: Based on this information, which of the following categories does this [item] belong to? Here are the [N] categories: ⟨labels⟩. Reply only one category that you think this [item] might belong to.

---

## F   QUANTITATIVE ANALYSIS OF SELECTED PSEUDO LABELS

To provide deeper insights into our *GNN-as-Judge* framework, we conduct a comprehensive analysis of the pseudo-label selection process across all datasets. This analysis examines the agreement patterns between LLM and GNN predictions, the accuracy distributions of selected labels, and the composition of the final augmented training sets. Table 5 presents the quantitative results of our pseudo-label selection strategy in the 3-shot setting.

### F.1   AGREEMENT AND DISAGREEMENT PATTERNS

Table 5: Quantitative analysis of pseudo label selection for the 3-shot setting.

| Dataset | Initial Pool | | | Final Set | | |
|---|---|---|---|---|---|---|
| | A/D Ratio | Agree Acc. | Disagree Acc. | Size | Final Acc. | Sel. Disagree Acc. |
| Cora | 978/520 | 93.35 | 61.68 | 1,103 | 92.11 | 82.41 |
| citeseer | 881/614 | 91.59 | 66.56 | 1,120 | 88.83 | 72.17 |
| Pubmed | 991/504 | 92.02 | 31.55 | 1,191 | 83.29 | 40.03 |
| ogbn-arxiv | 611/888 | 84.40 | 37.38 | 817 | 78.09 | 59.37 |
| ogbn-products | 753/733 | 87.12 | 37.24 | 883 | 81.99 | 52.29 |

Table 5 presents the quantitative results of our pseudo-label selection strategy in the 3-shot setting (seed=42). The table reveals several key patterns in how LLMs and GNNs agree or disagree on different datasets. The agreement/disagreement (A/D) ratios vary across datasets. Agreement nodes consistently achieve high accuracy, validating our theoretical analysis in Theorem 2. Notably, the disagreement accuracy is much lower and varies substantially across datasets, highlighting the importance of careful selection within the disagreement set.

Table 6: Accuracy (%) of our final selected pseudo-label set compared to baseline selection methods of the same size.

| Dataset | GNN Pred. Acc. | LLM Pred. Acc. | Sel. Acc. |
|---|---|---|---|
| Cora | 82.32 | 68.44 | **92.11** |
| citeseer | 81.07 | 60.80 | **88.83** |
| Pubmed | 71.20 | 82.61 | **83.29** |
| ogbn-arxiv | 55.93 | 50.55 | **78.09** |
| ogbn-products | 61.38 | 62.85 | **81.99** |

To further contextualize our framework's performance, Table 6 compares the accuracy of our final selected pseudo-label set "Sel. Acc") against baselines that randomly select a set of the same size using only GNN or only LLM predictions. The results clearly indicate that our method consistently outperforms both individual models across nearly all datasets.

| Dataset | Pearson | $\Delta_{L|G}$ | $\Delta_{G|L}$ |
|---|---|---|---|
| Cora | 0.258 | 0.216 | 0.162 |
| Citeseer | 0.202 | 0.159 | 0.115 |
| Pubmed | $-0.048$ | 0.020 | 0.062 |
| ogbn-arxiv | 0.117 | 0.047 | 0.058 |
| ogbn-products | 0.140 | 0.061 | 0.094 |

Table 7: Error independence diagnostics. $\Delta_{L|G} = P(\text{LLM err} \mid \text{GNN err}) - P(\text{LLM err})$ and $\Delta_{G|L} = P(\text{GNN err} \mid \text{LLM err}) - P(\text{GNN err})$. Smaller values indicate weaker dependence.

Table 8: Effect of preference score threshold $\tau$ on disagreement set size and GNN accuracy. Percentages denote proportion of the full disagreement set.

| Dataset | $\tau$ | Set Size | % of Full | Accuracy |
|---|---|---|---|---|
| Cora | 0.10 | 471 | 90.6% | 63.27% |
| | 0.40 | 262 | 50.4% | 75.95% |
| | 0.70 | 123 | 23.7% | 82.11% |
| | 0.90 | 35 | 6.7% | 82.86% |
| Citeseer | 0.10 | 586 | 95.4% | 67.64% |
| | 0.40 | 427 | 69.5% | 71.77% |
| | 0.70 | 238 | 38.8% | 72.27% |
| | 0.90 | 89 | 14.5% | 75.28% |
| Pubmed | 0.10 | 463 | 91.9% | 31.17% |
| | 0.40 | 325 | 64.5% | 36.63% |
| | 0.70 | 200 | 39.7% | 40.03% |
| | 0.90 | 51 | 10.1% | 52.94% |
| ogbn-arxiv | 0.10 | 815 | 91.8% | 39.37% |
| | 0.40 | 458 | 51.6% | 50.76% |
| | 0.70 | 205 | 23.1% | 59.51% |
| | 0.90 | 76 | 8.6% | 63.16% |
| ogbn-products | 0.10 | 625 | 85.3% | 40.22% |
| | 0.40 | 298 | 40.7% | 47.67% |
| | 0.70 | 130 | 17.7% | 52.31% |
| | 0.90 | 58 | 7.9% | 63.79% |

## F.2 ERROR CORRELATION ANALYSIS

We quantify error dependence using three complementary metrics:

- **Pearson correlation** between binary error indicators (1 = error), capturing linear dependence.
- $\Delta_{L|G} = P(L{=}1 \mid G{=}1) - P(L{=}1)$: the change in LLM error probability when the GNN errs. Positive values indicate the LLM is more likely to err when the GNN errs.
- $\Delta_{G|L} = P(G{=}1 \mid L{=}1) - P(G{=}1)$: the change in GNN error probability when the LLM errs.

Across all datasets, the dependence metrics remain small, indicating that GNN and LLM errors exhibit only weak correlation.

## F.3 DISAGREEMENT SET SIZE VS. QUALITY

To analyze the trade-off between disagreement set size and quality, we examine how different preference score thresholds ($\tau$) affect the filtered disagreement set. Table 8 shows the relationship between set size and GNN accuracy across datasets. Across all datasets, we observe a consistent pattern: higher filtering (smaller sets) yields substantially higher accuracy.

## F.4 SELECTION STRATEGY COMPARISON

The results in Table 9 reveal several important insights about pseudo label selection strategies. To ensure fair comparison, all methods follow the same experimental protocol: first selecting 1,500 candidate nodes using the respective selection strategy, then annotating them using Llama3-8B-

Table 9: Comparison of different pseudo label selection strategies on selected pseudo labels accuracy (%). **Best** results are bolded.

| Method | Cora | citeseer | ogbn-arxiv |
|---|---|---|---|
| Random | 64.44±3.23 | 60.80±2.15 | 53.93±1.87 |
| Degree | 65.88±1.09 | 64.26±1.89 | 50.24±2.86 |
| AGE | 67.36±1.74 | 65.78±0.94 | 52.31±1.45 |
| Confidence | 75.71±0.84 | **73.89±1.77** | **66.14±1.48** |
| LLM-GNN | **76.49±0.74** | 72.34±1.28 | 54.67±1.53 |

Instruct trained under 3-shot setting. The results demonstrate that traditional selection methods face significant limitations when LLMs are unreliable. Graph topology-based approaches achieve relatively low accuracy, indicating that structural information alone is insufficient for identifying nodes suitable for high-quality pseudo-labeling using LLMs. Interestingly, even LLM confidence-based selection cannot fully ensure high-quality pseudo labels. This highlights a fundamental challenge: when the underlying LLM is unreliable in low-resource settings, neither graph structure nor model confidence provides adequate signals for pseudo label selection. These findings motivate our *GNN-as-Judge* approach, which addresses these limitations through collaborative filtering that leverages both structural and semantic information.

## F.5 DIVERSITY ANALYSIS OF SELECTED NODES

To understand the diversity of nodes selected by our approach, we conduct an analysis to demonstrate the diversity and representativeness of nodes selected by our method. From Table 9, we observe that LLM confidence is a strong baseline for selecting high-quality pseudo labels. While pseudo label accuracy is crucial, selecting diverse nodes is equally important for effective LLM training, as it prevents overfitting to specific node types and improves generalization. We compare our approach with using LLM confidence to select the top 1,500 unlabeled nodes for the `ogbn-arxiv` dataset, extracting embeddings using Llama3-8B-Instruct.

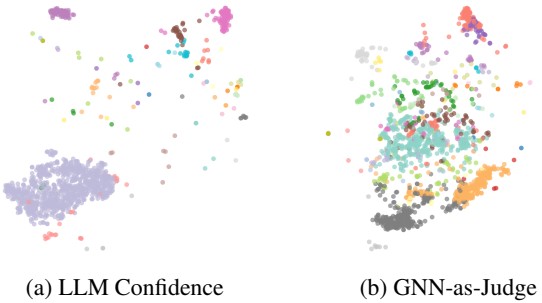

(a) LLM Confidence      (b) GNN-as-Judge

Figure 6: t-SNE visualization of selected pseudo-labeled nodes on the `ogbn-arxiv` dataset. Colors represent different classes.

Figure 6 provides a compelling visual comparison of the selection strategies. The LLM Confidence approach shows distinct, tight clusters with strong class separation, indicating it heavily favors nodes from a limited subset of the feature space. In contrast, our *GNN-as-Judge* approach selects nodes that are more broadly distributed across the entire feature space, covering a wider variety of node types and characteristics. This broader selection ensures that the LLM receives training examples from diverse areas of the data distribution, leading to better generalization across different node patterns.

Table 10 quantifies the diversity advantages of our approach. Coverage measures the average distance from each class centroid to the nearest selected node. Representativeness quantifies how well the selected nodes represent the overall data distribution using the Wasserstein distance between selected and full node distributions. Intra-class Variance measures the diversity within each class among selected nodes. This demonstrates that our approach selects nodes that are more representative of the overall data distribution.

Table 10: Quantitative diversity comparison between selection methods on `ogbn-arxiv` dataset.

| Method | Coverage ↓ | Representativeness ↓ | Intra-class ↑ |
|---|---|---|---|
| Confidence | 11.62 | 0.34 | 12.52 |
| **GNN-as-Judge** | **8.97** | **0.10** | **15.29** |

# G ADDITIONAL EXPERIMENTS

## G.1 SENSITIVITY TO $\lambda$

In our approach, the hyperparameter $\lambda$ controls the balance between standard instruction tuning and preference tuning as shown in Eq. 5. Figure 7 (left) illustrates how different $\lambda$ values affect performance on `Cora`, `citeseer` and `Pubmed` datasets in 3-shot settings. We observe a clear trend where lower $\lambda$ values yield significantly better results, with performance declining as $\lambda$ increases. This suggests that using a moderate weight for preference tuning loss will maintain better model performance.

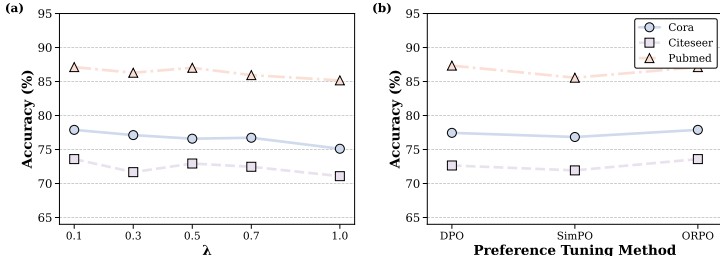

Figure 7: (a) Sensitivity analysis of the hyperparameter $\lambda$ showing performance trends across different values. Lower $\lambda$ values generally yield better performance, particularly for Cora and citeseer. (b) Performance comparison across different preference tuning losses (DPO, SimPO, ORPO), demonstrating the framework's compatibility with various optimization methods.

## G.2 ADAPTATION TO DIFFERENT PREFERENCE TUNING LOSSES

We evaluate the performance of our *GNN-as-Judge* framework across different preference tuning losses, comparing DPO (Rafailov et al., 2023), SimPO (Meng et al., 2024), and ORPO (Hong et al., 2024) on `Cora`, `citeseer` and `Pubmed` in 3-shot setting. As shown in Figure 7 (right), these preference tuning methods demonstrate relatively comparable performance. This observation highlights that our *GNN-as-Judge* is a plug-and-play framework where different preference tuning losses can be seamlessly integrated without compromising overall performance.

## G.3 EFFECTIVENESS OF INFLUENTIAL NODE SELECTION

To validate the importance of our influential node selection strategy, we compare the quality of pseudo labels generated for different node selection approaches. Specifically, we examine: (1) random selection of 1,000 unlabeled nodes, and (2) our influential node selection according to node incleunce score. Table 11 presents the pseudo-label accuracy when using either GNN or LLM predictions as pseudo labels.

## G.4 INTEGRATION WITH VARIOUS LLMS

To demonstrate the generalizability of our *GNN-as-Judge* framework, we evaluate its performance with different backbone LLMs. Table 12 presents results using Mistral-7B-Instruct (Jiang et al., 2023) and Llama3-8B-Instruct (Grattafiori et al., 2024) under the 3-shot setting, comparing their zero-shot baseline performance against our framework.

The results in Table 12 demonstrate the remarkable robustness of our *GNN-as-Judge* framework across diverse LLM families. Despite significant variations in the baseline zero-shot capabilities

Table 11: Comparison of pseudo-label accuracy (%) for different node selection strategies on 1,000 unlabeled nodes under 3-shot setting. Higher accuracy indicates better quality of selected nodes for pseudo-labeling.

| Dataset | Random Selection | | Influential Selection (Ours) | |
|---|---|---|---|---|
| | GNN Acc. | LLM Acc. | GNN Acc. | LLM Acc. |
| Cora | 68.24±1.83 | 66.92±2.14 | 82.39±1.26 | 68.84±1.52 |
| citeseer | 63.18±2.06 | 60.73±1.97 | 81.92±1.48 | 61.88±1.71 |
| Pubmed | 65.43±1.37 | 76.26±1.19 | 71.56±0.94 | 83.42±0.87 |
| ogbn-arxiv | 38.92±2.45 | 52.37±2.81 | 56.51±1.93 | 51.16±2.12 |
| ogbn-products | 60.15±1.94 | 73.83±1.76 | 64.54±1.41 | 62.91±1.28 |

Table 12: Performance comparison of different LLMs integrated with our GNN-as-Judge framework under 3-shot setting. Results show accuracy (%) with standard deviation.

| LLM | Method | Cora | citeseer | Pubmed | ogbn-arxiv | ogbn-products |
|---|---|---|---|---|---|---|
| Mistral-7B | Zero-Shot | 59.78±1.26 | 42.56±0.33 | 77.24±1.15 | 43.06±2.45 | 50.80±1.67 |
| | **GNN-as-Judge** | **76.16±1.15** | **73.08±0.89** | **84.79±0.94** | **57.36±1.52** | **79.02±1.41** |
| Llama3-8B | Zero-Shot | 65.54±0.26 | 58.17±0.64 | 74.51±0.39 | 50.18±1.44 | 75.48±1.54 |
| | **GNN-as-Judge** | **77.89±1.28** | **73.59±0.73** | **87.12±0.89** | **62.21±1.45** | **81.02±1.23** |

of these models, our framework consistently elevates performance to competitive levels across all architectures.

### G.5 PERFORMANCE ON HETEROPHILIC GRAPHS

To evaluate the effectiveness of our *GNN-as-Judge* framework on heterophilic graphs where connected nodes tend to have different labels, we conduct experiments on Cornell and Wisconsin (Wang et al., 2025) datasets using $H_2$GCN (Zhu et al., 2020) as the GNN backbone. These datasets exhibit low homophily ratios (0.11 for Cornell and 0.16 for Wisconsin), presenting unique challenges where traditional GNN assumptions of homophily do not hold. While our primary focus is not on

Table 13: Performance comparison on heterophilic graphs (Cornell and Wisconsin) using $H_2$GCN as GNN backbone under 3-shot setting. Results show accuracy (%) with standard deviation.

| Method | Cornell | Wisconsin |
|---|---|---|
| $H_2$GCN | 48.77±2.84 | 41.15±3.12 |
| Zero-Shot | 79.84±1.92 | 73.49±1.35 |
| **GNN-as-Judge** | **84.65±1.23** | **76.95±2.03** |

heterophilic graphs, these results demonstrate that our framework can seamlessly adapt to different GNN types without requiring architectural modifications. The consistent performance improvements over both the GNN baseline and zero-shot LLM approach validate the broad applicability of our collaborative pseudo-labeling strategy across diverse graph characteristics and GNN architectures.

### G.6 MEMORY USAGE ANALYSIS

Figure 8 presents a comprehensive comparison of the memory efficiency and accuracy trade-offs across different methods. The scatter plot illustrates the relationship between peak GPU memory consumption during training and achieved accuracy for each approach under 3-shot settings on the ogbn-arxiv dataset using a single NVIDIA A100 80GB GPU. LLM-based approaches require substantially higher memory usage. The analysis reveals a clear accuracy-memory trade-off, where methods achieving higher performance generally require more computational resources due to the integration of large language models.

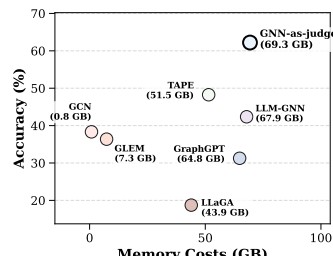

Figure 8: Memory usage versus accuracy.

## G.7 Integration with Different GNN Backbones

To assess the architectural robustness of the proposed framework, we evaluate its performance when paired with several widely used GNN backbones, including GCN, GAT, and GraphSAGE. Table 14 reports the performance across three representative datasets under 3-shot setting. The results show that the framework achieves consistent improvements regardless of the backbone used, suggesting that the method integrates well with a variety of GNN architectures.

Table 14: Performance of the proposed framework when integrated with different GNN backbones. Results are reported as mean accuracy $\pm$ standard deviation over three runs.

| Architecture | Citeseer | Cora | ogbn-arxiv |
|---|---|---|---|
| GCN | $73.59 \pm 0.64$ | $77.89 \pm 1.28$ | $71.24 \pm 0.42$ |
| GAT | $74.36 \pm 0.71$ | $79.12 \pm 0.62$ | $70.45 \pm 0.51$ |
| GraphSAGE | $74.71 \pm 1.09$ | $77.67 \pm 0.98$ | $69.82 \pm 1.04$ |

## H Limitations

While *GNN-as-Judge* demonstrates strong performance, several limitations exist. First, *GNN-as-Judge* incorporates several hyperparameters related to pseudo-label selection and loss weighting that require selection to achieve optimal performance. For future work, developing automatic methods to select the optimal number of pseudo-labels and reducing hyperparameter dependence could address these limitations. Additionally, updating the model concurrently with pseudo-label generation may further improve performance.

## I Extended Related Works

### I.1 Large Language Models Preference Alignment

Preference alignment refers to the process of aligning the outputs of language models with human preferences, often focusing on safety, helpfulness, and factuality (Askell et al., 2021; Ouyang et al., 2022). Reinforcement Learning from Human Feedback (RLHF) (Leike et al., 2018; Stiennon et al., 2020) is a prevalent method for aligning LMs with human preferences. The RLHF pipeline typically involves collecting human preference data, training a reward model, and using reinforcement learning to optimize the language model against this reward function (Bai et al., 2022; Christiano et al., 2017).

While RLHF has proven effective, its reliance on a separate reward model introduces computational and methodological complexities. To address these challenges, Direct Preference Optimization (DPO) (Rafailov et al., 2023) has emerged as a more efficient alternative, eliminating the need for an explicit reward model by directly optimizing preference probabilities. Several extensions and refinements of DPO have since been proposed. For instance, KTO (Ethayarajh et al., 2024) incorporates insights from prospect theory to enhance preference learning. Other variants, such as GPO (Zhao et al., 2024), $\Psi$PO (Azar et al., 2024), and ODPO (Amini et al., 2024), further improve or generalize DPO in different theoretical and practical aspects. Recent advancements, including ORPO (Hong et al., 2024) and SimPO (Meng et al., 2024), simplify the alignment pipeline by removing the need for a reference model while maintaining competitive performance. Despite these advancements, the application and adaptation of preference alignment techniques to structured tasks such as graph-based learning remain underexplored.

## J The Use of Large Language Models (LLMs)

In accordance with transparency and reproducibility standards, we disclose that LLMs were employed to assist with the preparation of this manuscript. Specifically, we utilized LLMs for language polishing, grammatical correction, and stylistic improvements to enhance the clarity and readability of our technical writing.

