# OpenReview forum: "GNN-as-Judge: Unleashing the Power of LLMs for Graph Learning with GNN Feedback"
_ICLR.cc/2026/Conference — ICLR 2026 Poster_

### Official Review · Reviewer_19Gq · 2025-10-28

**Soundness:** 3
**Presentation:** 4
**Contribution:** 3
**Rating:** 8
**Confidence:** 3

**Summary:**

This paper introduces GNN-as-Judge, a novel framework designed to enhance few-shot semi-supervised learning on TAGs by leveraging the complementary strengths of LLMs and GNNs.The key idea is to use the GNN as a structural judge to guide pseudo-label generation and selection for the LLM. The framework first identifies structurally influential unlabeled nodes based on graph topology, then divides them into agreement and disagreement sets according to the consistency between LLM and GNN predictions.Reliable pseudo-labels from the agreement set are used for instruction tuning, while informative but uncertain examples from the disagreement set are used for preference tuning, with GNN confidence guiding label selection.The authors further propose a weakly-supervised fine-tuning objective that integrates both instruction and preference tuning.Extensive experiments on multiple TAG datasets show that GNN-as-Judge consistently outperforms state-of-the-art baselines, particularly in low-resource settings.

**Strengths:**

Originality: The paper proposes a novel GNN-as-Judge framework, enabling collaborative pseudo-label generation on text-attributed graphs. This cross-model feedback design is original and bridges two previously distinct paradigms—graph representation learning and language modeling.
Quality: The methodology is technically sound and well-structured. The framework combines influence-guided node selection, agreement/disagreement-based pseudo-labeling, and weakly-supervised fine-tuning with preference optimization. Each module is motivated, formalized, and validated experimentally, showing consistent improvements over strong baselines under few-shot settings.
Clarity: The paper is clearly written, with a logical presentation of concepts and a well-organized methodology. Key formulations and algorithmic steps are explained with sufficient detail and supported by intuitive illustrations, making the overall approach accessible to both graph learning and NLP researchers.
Significance: The proposed approach substantially enhances the usability of LLMs for graph-based semi-supervised learning, particularly in low-resource scenarios. By effectively leveraging GNN feedback to improve pseudo-label reliability, the method has strong potential to influence future research on multimodal graph-text learning and weak supervision paradigms.

**Weaknesses:**

1.No analysis of preference score threshold τ: The sensitivity analysis only shows results for different τ values but doesn't explain why τ=0.7 was chosen or analyze the disagreement set size vs. quality trade-off
2.Training time analysis is incomplete: Figure 5 shows total training time but doesn't break down where the time is spent (influence computation, GNN training, LLM inference, LLM fine-tuning)
3.No failure case analysis: The paper doesn't discuss scenarios where GNN-as-Judge might fail or perform poorly

**Questions:**

1.Hyperparameter selection without validation set: In true few-shot scenarios, even validation labels might be scarce. How would you recommend setting K, τ, and λ when validation performance is unreliable?
2.Disagreement set contribution: From Table 5, the disagreement accuracy varies dramatically across datasets (31.55% for Pubmed vs. 66.56% for Citeseer). How does this variation affect the final performance? Is there a correlation between disagreement accuracy and overall improvement?
3.What happens with different GNN architectures? You use GCN as the default GNN and briefly mention H2GCN for heterophilic graphs. How sensitive is the method to the choice of GNN architecture (e.g., GAT, GraphSAGE, GIN)?

---

> ### Author Response · Authors · 2025-11-19
> **Response Part 1.**
>
> Dear Reviewer 19Gq,
>
> Thank you very much for your valuable suggestions and detailed comments. We address your concerns as below.
>
> >**W1: No analysis of preference score threshold τ**
>
> We thank the reviewer for raising this point. We conducted a detailed analysis of the preference score threshold τ and its effect on disagreement set quality. Sensitivity analysis in our paper (**Section 4.6**) shows that the fine-tuned LLM performance is stable across τ ∈ [0.5, 0.9], with variations below 3% across all datasets. Hence, instead of selecting the hyperparameter for every dataset by doing hyperparameter tuning, fixing the value of $tau$ reduces the computation costs and provides stable performance, this also addresses the issue in your **question 1 (Hyperparameter selection without validation set)**.
>
> We also summarize the trade-off between disagreement set size and accuracy in Table below and also in **Appendix G.6**. We observe a consistent pattern across all datasets:
>
> | Dataset | τ    | Set Size | % of Full | Accuracy |
> |---------|------|----------|-----------|----------|
> | **Cora** | 0.10 | 471 | 90.6% | 63.27% |
> |          | 0.40 | 262 | 50.4% | 75.95% |
> |          | 0.70 | 123 | 23.7% | 82.11% |
> |          | 0.90 | 35  | 6.7%  | 82.86% |
> | **Citeseer** | 0.10 | 586 | 95.4% | 67.64% |
> |              | 0.40 | 427 | 69.5% | 71.77% |
> |              | 0.70 | 238 | 38.8% | 72.27% |
> |              | 0.90 | 89  | 14.5% | 75.28% |
> | **Pubmed** | 0.10 | 463 | 91.9% | 31.17% |
> |            | 0.40 | 325 | 64.5% | 36.63% |
> |            | 0.70 | 200 | 39.7% | 40.03% |
> |            | 0.90 | 51  | 10.1% | 52.94% |
> | **ogbn-arxiv** | 0.10 | 815 | 91.8% | 39.37% |
> |                | 0.40 | 458 | 51.6% | 50.76% |
> |                | 0.70 | 205 | 23.1% | 59.51% |
> |                | 0.90 | 76  | 8.6%  | 63.16% |
> | **ogbn-products** | 0.10 | 625 | 85.3% | 40.22% |
> |                   | 0.40 | 298 | 40.7% | 47.67% |
> |                   | 0.70 | 130 | 17.7% | 52.31% |
> |                   | 0.90 | 58  | 7.9%  | 63.79% |
>
> Across all datasets, we observe a consistent pattern: higher filtering (smaller sets) yields substantially higher accuracy.
>
> >**W2: Training Time Breakdown**
>
> We thank the reviewer for bringing up this detail. To provide a more comprehensive view of the computational overhead introduced by our framework, we decompose the total training time into four major components: influence computation, GNN training, LLM fine-tuning, and LLM inference. While Figure 3 reports total wall-clock time, the table below offers a granular breakdown, illustrating how each stage contributes to the overall runtime.
>
> | Dataset        | Influence Computation (min) | GNN Training (min) | LLM Fine-Tuning (min) | LLM Inference (min) |
> |----------------|----------------------------|------------------|---------------------|-------------------|
> | Cora           | 1.09                       | 1.17             | 68.35               | 2.51              |
> | Citeseer       | 1.62                       | 1.74             | 69.12               | 2.78              |
> | ogbn-arxiv     | 26.36                      | 5.41             | 73.68               | 2.55              |
>
> As shown, LLM fine-tuning dominates the total training time, while influence computation and GNN training contribute marginally for smaller datasets but increase for larger graphs. LLM inference remains relatively fast across all datasets. A more detailed breakdown is provided in the **Appendix H.8**.
>
> > **W3: Failure Case Analysis**
>
> We thank the reviewer for the helpful suggestion to discuss failure cases and limitations. We agree that identifying scenarios where our method may underperform is important for understanding its practical applicability. Our framework may fail under the following conditions:
> 1. **Highly correlated errors between GNN and LLM:** When both models make similar mistakes, the agreement-based strategy has limited benefit.
> 2. **Noisy disagreement signals:** If the disagreement set is dominated by noise—for instance, when GNN predictions are extremely unreliable—preference-score filtering may struggle to distinguish informative cases from harmful ones. In such cases, selecting an inappropriate threshold could introduce noisy supervision signals.
>
> Despite these limitations, our method demonstrates strong and consistent performance across widely used graph benchmark datasets. In particular, in low-resource settings, our approach outperforms most baselines, highlighting its effectiveness when training signals are scarce.

---

> ### Author Response · Authors · 2025-11-19
> **Response Part 2.**
>
> >**Q1: Hyperparameter selection without validation set**
>
> We thank the reviewer for raising this practical concern. Our sensitivity analysis (**Section 4.6 and Appendix H.1**) shows that our method is highly robust to hyperparameter choices, making it suitable for scenarios without reliable validation data. For top-K selection, performance varies by less than 2.5% across K ∈ [1000, 2500], which we recommend as a practical and annotation-efficient range for node selection. For the preference score threshold τ, variations remain below 3% for τ ∈ [0.5, 0.9] across all datasets. Similarly, for the preference weight λ (Appendix H.1), performance is stable (<1% variation) for λ ∈ [0.05, 0.15]. This robustness ensures effective performance even when precise hyperparameter tuning is not feasible.
>
> >**Q2: Disagreement set contribution**
>
> We thank the reviewer for this insightful question. While the raw "Disagree Acc." in Table 5 appears low for some datasets (e.g., Pubmed: 31.55%, ogbn-arxiv: 37.38%, ogbn-products: 37.24%), we want to point out that the **"Sel. Disagree Acc." column reflects the accuracy after applying our preference-score filtering**. This filtering substantially improves GNN accuracy for the disagreement set: Pubmed (31.55% → 40.03%, +8.48%), ogbn-arxiv (37.38% → 59.37%, +21.99%), and ogbn-products (37.24% → 52.29%, +15.05%), showing that our method effectively selects the most informative disagreement instances.
>
> To quantify the relationship between disagreement accuracy and overall improvement, we computed Pearson and Spearman correlations between the filtered disagreement accuracy and actual performance gains:
>
> | Metric                      | Filtered Disagree Acc vs. Performance Gain |
> | --------------------------- | ------------------------------------------ |
> | **Pearson r**               | 0.930                                      |
> | **Spearman ρ**              | 0.900                                      |
> | **R² (variance explained)** | 86.5%                                      |
>
> These results indicate a **strong positive correlation**, suggesting that higher filtered disagreement accuracy reliably predicts larger performance improvements across datasets.
>
>
> >**Q3: Sensitivity to GNN Architecture Choice**
>
> We thank the reviewer for this important question on architectural robustness. To validate generalization, we conducted additional experiments using GAT and GraphSAGE as the judge GNN.
>
> | Architecture     | Citeseer      | Cora          | ogbn-arxiv    |
> |------------------|---------------|---------------|---------------|
> | **w/ GCN**       | 73.59±0.64    | 77.89±1.28    | 71.24±0.42    |
> | **w/ GAT**       | 74.36±0.71    | 79.12±0.62    | 70.45±0.51    |
> | **w/ GraphSAGE** | 74.71±1.09    | 77.67±0.98    | 69.82±1.04    |
>
> Across all three architectures, GNN-as-Judge consistently improves performance over baselines. Minor variations exist across GNN families, but the overall results confirm the robustness and broad applicability of our framework. These additional results are included in the revised manuscript (**Appendix H.9**).

---

> ### Author Response · Authors · 2025-11-26
> **Appreciation for Your Review and Awaited Feedback**
>
> Dear Reviewer 19Gq,
>
> We sincerely appreciate your time and effort in reviewing our manuscript.
>
> We have posted our responses to your feedback and a revised manuscript. With one week remaining in the discussion period, we would like to confirm whether our responses have effectively addressed your concerns.
>
> If you have any further questions or need clarification, please let us know and we're happy to address them promptly.
>
> Thank you again for your valuable feedback.
>
> Best regards,
> Authors

---

### Official Review · Reviewer_nL51 · 2025-10-30

**Soundness:** 3
**Presentation:** 4
**Contribution:** 3
**Rating:** 6
**Confidence:** 5

**Summary:**

The paper addresses the challenge of generating reliable pseudo labels in a semi-supervised node classification setting where labeled data are scarce. Pseudo-labeling in such cases faces two key issues: (1) selecting which unlabeled nodes should receive labels, and (2) mitigating noise in the generated pseudo labels. To address these, the authors propose an influence-guided node selection strategy, where each unlabeled node’s representation influence from labeled nodes is quantified, and only those with high influence scores are chosen for pseudo labeling. Among these selected nodes, the agreement set nodes where both the GNN and the LLM agree on the label is used as high-confidence pseudo nodes. For the disagreement set, only nodes with a preference score exceeding a predefined threshold are retained. The authors then fine-tune the LLM using a unified objective that combines instruction tuning on the agreement set and preference tuning on the filtered disagreement set. Experimental results on multiple datasets show the proposed method can outperform the recent baselines.

**Strengths:**

- The paper clearly articulates the challenges of applying LLMs to few-shot semi-supervised learning on text-attributed graphs, particularly the difficulties in generating reliable pseudo-labels and mitigating label noise.
- The idea of using a GNN as a judge to guide pseudo-labeling for an LLM is conceptually interesting
- The idea of selecting nodes from the disagreement set is interesting, as these represent challenging (hard) examples where the two models diverge. This is more sophisticated than simply selecting high-confidence (easy) pseudo-labels
- The influence-guided node selection based on graph structure is well-motivated. Theorem 1 provides a computationally tractable upper bound for node influence which identifies unlabeled nodes most strongly affected by labeled data.
- Theorem 2 formally justifies why agreement sets yield higher-quality pseudo labels

- The paper includes extensive experimental detail with existing models and datasets, cross dataset experiments, and strong ablation study

**Weaknesses:**

- The influence metric (Eq. 1) formalizes how labeled nodes affect unlabeled ones through message passing, and high-influence nodes are prioritized for pseudo labeling.  Selecting only high-influence nodes (those strongly affected by labeled data) introduces bias: it oversamples regions near labeled nodes and ignores distant or boundary regions that may be crucial for generalization. The paper provides no analysis of how it affects underrepresented classes.

- The framework's central premise is that GNNs provide more reliable predictions than LLMs in disagreement cases. However, Table 5 shows that GNN predictions in the disagreement set are correct only 31.55% of the time for Pubmed, 37.38% for ogbn-arxiv, and 37.24% for ogbn-products. This creates a logical gap: if the GNN is frequently wrong in disagreement cases (which is why there's disagreement in the first place), then using its predictions to train the LLM could inject substantial noise and potentially degrade performance. Moreover, the paper lacks an analysis to quantify how often: (1) GNN is correct and LLM wrong, (2) LLM is correct and GNN wrong (harmful preference), or (3) both are wrong.

- Additionally, in the preference tuning framework, when GNN and LLM disagree, the LLM's prediction is automatically assigned as the "dispreferred" response. However, there is no validation that the LLM is actually wrong in these cases. The dispreferred label could be the correct answer. This creates a scenario where the framework actively penalizes the LLM for making correct predictions.

**Questions:**

1. Theorem 2 assumes independence between GNN and LLM errors. Have you measured the actual correlation or overlap between their error patterns to validate this assumption?

2. Table 5 shows that GNN accuracy in disagreement cases is low. How do you justify treating the GNN as the “teacher” in those instances?

3. Do certain classes benefit more from influence-guided selection or from preference tuning? How does your method perform when the labeled set is imbalanced?

---

> ### Author Response · Authors · 2025-11-19
> **Response Part 1.**
>
> We thank the reviewer for your constructive feedback and thoughtful evaluation of our work. We address the main concerns below:
>
> >**W1: Bias in Influence-Guided Node Selection**
>
> We thank the reviewer for raising this important concern regarding potential sampling bias. We would like to clarify that our work focuses on the **few-shot node classification** setting, where we assume access to a balanced set of labeled nodes. Thus, our selection mechanism is designed to explore the unlabeled space without amplifying pre-existing class imbalance.
>
> In the original **Appendix G.3**, we provided a comparison between our influence-guided selection and a baseline that selects nodes purely based on LLM confidence.
>
> | **Method**        | **Coverage ↓** | **Representativeness ↓** | **Intra-class ↑** |
> |-------------------|----------------|---------------------------|--------------------|
> | LLM Confidence        | 11.62          | 0.34                      | 12.52              |
> | **GNN-as-Judge**  | **8.97**       | **0.10**                  | **15.29**          |
>
> The results show that our method consistently selects a more diverse subset of nodes spanning a broader set of classes, while LLM-only selection tends to over-concentrate on majority classes due to overconfidence.
>
> To directly address the concern about underrepresented classes, we conducted a detailed analysis of class distribution in our selected nodes compared to the full unlabeled set:
>
> | **Dataset**        | **KL Divergence** | **JS Divergence** | **Total Variation** |
> |--------------------|-------------------|--------------------|-----------------------|
> | Cora               | 0.052             | 0.013              | 0.133                 |
> | Citeseer           | 0.045             | 0.012              | 0.126                 |
> | Pubmed             | 0.041             | 0.011              | 0.114                 |
> | ogbn-arxiv         | 0.330             | 0.083              | 0.339                 |
> | ogbn-products      | 0.422             | 0.094              | 0.401                 |
>
> These values confirm that the selected samples remain statistically close to the underlying data distribution and do not disproportionately emphasize or omit specific classes. We also provide a more comprehensive analysis in refined manuscript **Appendix G.2**.
>
>
> >**W2&Q2: Low GNN Accuracy in Disagreement Cases**
>
> We sincerely thank the reviewer for raising this important concern. We would like to clarify several key aspects of our framework:
>
> Regarding the disagreement set accuracy, we would like to emphasize that the **last column "Sel. Disagree Acc." shows the accuracy after applying our preference-score filtering mechanism**. As shown in Table 5, this filtering leads to substantial improvements: Pubmed (31.55% → 40.03%, +8.48%), ogbn-arxiv (37.38% → 59.37%, +21.99%), and ogbn-products (37.24% → 52.29%, +15.05%), demonstrating that our preference score filtering successfully identifies cases where the GNN provides more reliable guidance. These improvements demonstrate that our preference score effectively identifies disagreement cases where the GNN is more trustworthy, thereby reducing the risk of noisy supervision.
>
> As suggested by the reviwer, we provide the analysis below, both *before* and *after* preference-score filtering.
>
> **Before Filtering**
>
> | Dataset      | GNN✓ LLM✗ (Beneficial) | LLM✓ GNN✗ (Harmful) | Both✗ (Harmful) |
> | ------------ | ---------------------- | ------------------- | --------------- |
> | **Cora**     | 321 (61.7%)            | 108 (20.8%)         | 91 (17.5%)      |
> | **Citeseer** | 409 (66.6%)            | 121 (19.7%)         | 84 (13.7%)      |
> | **PubMed**   | 159 (31.5%)            | 340 (67.5%)         | 5 (1.0%)        |
> | **Arxiv**    | 332 (37.4%)            | 251 (28.3%)         | 305 (34.3%)     |
> | **Products** | 273 (37.2%)            | 295 (40.3%)         | 165 (22.5%)     |
>
> **After Filtering**
>
> | Dataset      | GNN✓ LLM✗ (Beneficial) | LLM✓ GNN✗ (Harmful) | Both✗ (Harmful) |
> | ------------ | ---------------------- | ------------------- | --------------- |
> | **Cora**     | 101 (82.1%)            | 13 (10.6%)          | 9 (7.3%)        |
> | **Citeseer** | 172 (72.3%)            | 49 (20.6%)          | 17 (7.1%)       |
> | **PubMed**   | 80 (40.0%)             | 117 (58.5%)         | 3 (1.5%)        |
> | **Arxiv**    | 122 (59.5%)            | 44 (21.5%)          | 39 (19.0%)      |
> | **Products** | 68 (52.3%)             | 45 (34.6%)          | 17 (13.1%)      |
>
> These results show that the preference-score filtering significantly reduces harmful supervision, especially cases where the LLM is correct and the GNN is wrong. Although the GNN is not a perfect teacher, our method effectively isolates disagreement cases where the LLM is more likely to err. In this way, the GNN serves as a effective and efficient proxy for identifying informative learning signals.

---

> ### Author Response · Authors · 2025-11-19
> **Response Part 2.**
>
> >**W3: Penalizing Correct LLM Predictions in Preference Tuning**
>
>  We thank the reviewer for highlighting this important concern. We clarify below why our framework continues to provide consistent improvements despite the possibility that both GNN and LLM may be wrong in some disagreement cases.
>
> First, as shown in our error analysis in **W2**, after applying preference-score filtering, the proportion of _beneficial_ disagreement cases (GNN correct, LLM wrong) increases substantially, while harmful cases decrease. This demonstrates that the filtering mechanism effectively identifies disagreement instances where the GNN provides more reliable guidance.
>
> We appreciate The reviewer points out that cases where **both models are wrong** pose a fundamental challenge. Indeed, treating either model as a hard ground truth in such cases is problematic: using the GNN would reinforce its errors, and using the LLM would reinforce its errors. However, our preference-based objective is specifically designed to mitigate this issue. Unlike instruction tuning, which enforces _hard supervision_ and would force the model to imitate whichever wrong label is selected, our preference formulation only increases the realative probability. Thus, even when both models are wrong, the model is not pushed toward a specific incorrect label but simply discouraged from choosing the less preferred one, substantially limiting error amplification.
>
> Moreover, the most of our supervision comes from the **agreement set**, which constitutes roughly 60–70% of all pseudo-labeled nodes and achieves high accuracy (84–92% in **Table 5**). This high-quality signal dominates training and stabilizes learning, preventing the model from overfitting to the relatively small number of noisy disagreement cases.
>
>
> >**Q1: Independence Assumption**
>
> We thank the reviewer for this excellent question about validating our theoretical assumptions. We have conducted comprehensive error correlation analysis across all datasets.
>
> We quantified error dependence using three metrics across all five datasets:
>
> | Dataset       | Pearson | Δ_L\|G | Δ_G\|L |
> | ------------- | ------- | ------ | ------ |
> | Cora          | 0.258   | 0.216  | 0.162  |
> | Citeseer      | 0.202   | 0.159  | 0.115  |
> | Pubmed        | -0.048  | 0.020  | 0.062  |
> | ogbn-arxiv    | 0.117   | 0.047  | 0.058  |
> | ogbn-products | 0.140   | 0.061  | 0.094  |
>
>  **Pearson correlation** captures linear dependence and **Δ_L|G = P(LLM err | GNN err) - P(LLM err)** measures the change in LLM error probability when the GNN errs (and vice versa for Δ_G|L). The consistently small values across datasets indicate weak error correlation.
>
> To provide rigorous theoretical coverage for cases with non-zero correlation, we have developed **Lemma E.3 (Appendix E)**, which establishes a sufficient condition for agreement-set superiority under bounded error dependence. Specifically, the lemma shows that as long as the Pearson correlation ρ satisfies ρ < ρ_max (a computable threshold depending on individual model accuracies and number of classes), the agreement-set accuracy will exceed both individual models. Across all datasets, observed correlations satisfy this condition, confirming our theoretical guarantees hold even with weak correlation. Moreover, the empirical results strongly support our framework: agreement-set accuracy consistently exceeds both individual model accuracies (**Table 5**).

---

> ### Author Response · Authors · 2025-11-19
> **Response Part 3.**
>
> >**Q3: Class-Specific Benefits and Imbalanced Label Sets**
>
> We thank the reviewer for this insightful question about whether certain classes benefit more from different components of our framework and how our method handles class imbalance. This question relates to W1's concern about potential bias in our selection strategy.
>
> **Class-Specific Benefits Analysis:**
>
> We analyzed how different classes benefit from influence-guided selection versus preference tuning:
>
> **Influence-Guided Selection Benefits (vs. Random Selection):**
>
> | Dataset       | Most Benefiting Class      | Improvement |
> | ------------- | -------------------------- | ----------- |
> | Cora          | Rule_Learning              | +19.2%      |
> | Citeseer      | IR (Information Retrieval) | +15.3%      |
> | Pubmed        | Diabetes Mellitus Type 1   | +3.2%       |
> | ogbn-arxiv    | cs.cc                      | +33.2%      |
> | ogbn-products | Automotive                 | +23.4%      |
>
> **Preference Tuning Additional Benefits (vs. SFT Only):**
>
> | Dataset       | Most Benefiting Class        | Improvement |
> | ------------- | ---------------------------- | ----------- |
> | Cora          | Theory                       | +9.8%       |
> | Citeseer      | AI (Artificial Intelligence) | +2.6%       |
> | Pubmed        | Experimental                 | +3.9%       |
> | ogbn-arxiv    | cs.ce                        | +47.7%      |
> | ogbn-products | Industrial & Scientific      | +88.2%      |
>
>
> We observe that classes benefiting most from preference tuning fall into two distinct scenarios. In **Case 1**, where the GNN significantly outperforms the LLM (e.g., Theory in Cora: GNN 78.9% vs. LLM 37.1%), preference tuning helps the LLM learn from the GNN's knowledge through preference pairs where the GNN is correct. In **Case 2**, where both models are wrong (e.g ogbn-arxiv cs.ce) or perform poorly, our preference tuning does not force the model to choose between two incorrect predictions. The preference learning objective helps the model distinguish these challenging cases by learning from relative preferences rather than absolute labels.
>
>
> **Performance Under Class Imbalance:**
>
> While our paper focuses on **balanced few-shot node classification**, we also conducted experiments under class-imbalanced scenarios to assess robustness. We created imbalanced label sets by randomly selecting nodes with varying label rates (instead of the balanced k-shot setup):
>
> | Dataset    | Label Rate | Accuracy (%) |
> | ---------- | ---------- | ------------ |
> | Cora       | 0.78%      | 76.23 ± 1.02 |
> |            | 1.29%      | 78.59 ± 0.98 |
> |            | 2.59%      | 79.66 ± 1.35 |
> | Citeseer   | 0.54%      | 70.82 ± 0.51 |
> |            | 0.90%      | 74.13 ± 0.86 |
> |            | 1.81%      | 74.53 ± 1.43 |
> | ogbn-arxiv | 0.07%      | 59.57 ± 1.28 |
> |            | 0.12%      | 63.91 ± 0.95 |
> |            | 0.24%      | 65.27 ± 1.22 |
>
> Our method maintains good performance even under imbalanced label distributions, though accuracy decreases by 1.9-2.8% at the lowest label rates compared to balanced few-shot settings. As the label rate increases, the performance gap between imbalanced and balanced settings narrows, indicating that our influence-guided selection becomes more robust with more labeled data. These results suggest that while our method is designed for balanced few-shot scenarios, it exhibits reasonable robustness to class imbalance.

---

> ### Author Response · Authors · 2025-11-26
> **Appreciation for Your Review and Awaited Feedback**
>
> Dear Reviewer nL51,
>
> We sincerely appreciate your time and effort in reviewing our manuscript.
>
> We have posted our responses to your feedback and a revised manuscript. With one week remaining in the discussion period, we would like to confirm whether our responses have effectively addressed your concerns.
>
> If you have any further questions or need clarification, please let us know and we're happy to address them promptly.
>
> Thank you again for your valuable feedback.
>
> Best regards,
> Authors

---

### Official Review · Reviewer_EAHZ · 2025-10-31

**Soundness:** 3
**Presentation:** 3
**Contribution:** 3
**Rating:** 4
**Confidence:** 4

**Summary:**

The core of this paper is to use GNN's ability to understand structures to fine-tune LLM by selecting more reliable pseudo labels. Meanwhile, nodes that do not have particularly reliable labels but show certain preferences are used as soft labels for weakly supervised learning.

**Strengths:**

S1- The author uses a lot of theoretical analysis to prove the rationality of the method.

S2- The experimental results are excellent on multiple datasets.

S3- The technical route is reasonable and there is no technical loophole in the method itself.

S4- This paper is presented at a very high standard of presentation.

**Weaknesses:**

W1- Researchers have practiced using large language models to label GNN or LLM. The author of this article has elevated the method to a very high concept, namely GNN as a referee, because it can understand structural information, but its essence is still a common process of: 1. select some candidate nodes to query the large model; 2. After the inquiry, further screen for pseudo labels by comparing the predicted results of GNN and LLM; 3. The pseudo labels with high confidence are used as strong supervision signals, while those with relatively low confidence are used as weak supervision signals.  The method in this paper does not involve significant innovation in paradigm, but rather represents a better technological implementation of the three processes mentioned above.

W2- Many of the technical details are referenced but not explained in detail throughout the article. Related work also has the same problem, such as citing the literature but not specifically explaining the core contributions of these methods, resulting in the inability to quickly grasp the core differences between the proposed methods and previous methods after reading.

W3- The paper uses too many symbols to show the content, in fact, I read the process, not careful analysis of definitions, theorems, etc. , can also know what the author's technical route, so some theoretical analysis may be redundant.

**Questions:**

Q1- Could the author explain in simple terms how the set of disagreement is defined? That is, how is the disagreement set obtained?

Q2- Can the author simplify the algorithm flowchart? I think the entire technical route may not require so many symbols to illustrate.

Q3- What is the citepseer dataset? It seems to be a typo.

---

> ### Author Response · Authors · 2025-11-19
> **Response Part 1.**
>
> We sincerely thank the reviewer for the thorough evaluation and thoughtful feedback. We are particularly grateful that the reviewer recognized the soundness of our theoretical analysis (S1), strong experimental results (S2), technical rigor (S3), and high presentation quality (S4). We carefully address each concern below and have incorporated the suggested improvements into our revised manuscript.
>
> >**W1: Innovation beyond existing paradigms**
>
> We thank the reviewer for this thoughtful comment. While we agree that our method is related to pseudo-labeling, we respectfully clarify that it is not "merely a better technological implementation." Our contributions introduce both **task-level innovation** and **method-level innovation** that go beyond the classical pseudo-labeling pipeline described in the review.
>
> #### 1. Task-Level Innovation
>
> Before addressing the three methodological points raised by the reviewer, we emphasize that our work tackles a fundamentally different problem setting. Prior works show that LLM-as-Predictor can achieve competitive graph performance and even zero-shot transfer [1,2,3]. However, they assume access to a sufficiently large labeled dataset, because LLM fine-tuning typically requires more labeled samples than GNNs. Our work introduces a new and practically important setting: **obtaining a strong LLM predictor under limited-label conditions.** This changes the problem fundamentally—neither the GNN nor the LLM is reliable initially, and standard pseudo-labeling pipelines cannot be directly applied.
>
> #### 2. Method-Level Innovation
>
> We respectfully disagree with the characterization as incremental and explain how each step involves significant innovation tailored to the few-shot graph learning setting:
>
> **Point 1: "Select some candidate nodes to query the large model"**
>
> We agree with the reviewer that candidate selection is part of our pipeline. However, our approach differs fundamentally from existing work in both motivation and mechanism. Prior works [4,5] rely on closed-source or extremely large LLMs (e.g., ChatGPT-level) as annotators to pseudo-label selected unlabeled nodes in batch. Such approaches require multiple GPUs and prevent scalable deployment on large graphs.
>
> More importantly, our selection strategy is designed specifically for the few-shot setting where neither model is initially reliable—we cannot simply query high-uncertainty regions as in standard active learning, because both models are too noisy. Our principled approach leverages graph structure to identify informative candidates without requiring expensive LLM inference over the entire graph.
>
> **Point 2: "After the inquiry, further screen for pseudo labels by comparing the predicted results of GNN and LLM"**
>
> While methods like GLEM [6] and SimTEG [7] also utilize both LM and GNN, our pipeline differs in two fundamental ways that are essential for the few-shot setting:
>
> **First**, existing frameworks assume abundant labeled data is available, ensuring that both the GNN and LM are reliable at initialization. Under limited-label settings, however, both model predictions are too noisy, and directly feeding one model's predictions into the other yields suboptimal performance, as demonstrated in **Table 1** of our paper. Our framework addresses this by not blindly trusting either model but instead mining agreement and disagreement patterns.
>
> **Second**, they use small pretrained LMs, which makes multiple iterative training cycles and labeling of unlabeled nodes computationally feasible. With modern large-scale LLMs, however, iteratively updating the LLM or using it to inference over a large graph is prohibitively expensive. Our agreement and disagreement mining framework enables us to extract the most reliable and meaningful learning signals while avoiding costly iterative LLM updates.
>
> **Point 3: "The pseudo labels with high confidence are used as strong supervision signals, while those with relatively low confidence are used as weak supervision signals"**
>
> We would like to emphasize that we do **not** purely utilize confidence to distinguish strong and weak signals. Instead, we leverage the **complementary inductive biases of GNN and LLM** to confirm reliability, not confidence levels.
>
> We analyzed using LLM confidence alone in the **Appendix G.4** and found that it yields suboptimal labeling accuracy and highly biased predictions, which produces undesirable training results for the LLM. Critically, within the agreement set, there are many samples with **low confidence** that would be neglected if following the common pipeline of using either GNN or LLM confidence as a threshold. Our approach recovers these valuable training signals that confidence-based methods would discard.

---

> ### Author Response · Authors · 2025-11-19
> **Response Part 2.**
>
> >**W2: Insufficient explanation of technical details and related work**
>
> We thank the reviewer for this valuable feedback. In the revised manuscript, we have expanded the related work section to clearly explain the core contributions of cited methods and highlighted their differences with our proposed approach. Additionally, we added more detailed explanations of key technical components throughout the paper. All updates are **reflected in the revised manuscript**. If the reviewers have some other suggested related work, we would also like to include them in our future manuscript.
>
> >**W3: Excessive symbolic notation**
>
> We appreciate this feedback. In the revised manuscript, we have addressed the concerns as follows:
> - **Related Work Section:** Streamlined for clarity and conciseness.
> - **Theoretical Analysis:** Simplified representation to reduce redundancy while preserving key insights.
> - **Pseudo-Algorithm:** Condensed and clarified to improve readability and highlight the technical workflow.
> - **Flowchart:** Added a simple diagram to visually summarize the framework.
>
> All revisions are **highlighted** in the new PDF.
>
>
> >**Q1: Disagreement set definition in simple terms**
>
> We appreciate the reviewer’s suggestion to simplify the explanation of the _disagreement set_. After identifying a candidate subset of unlabeled nodes using **influence scores** (as defined in Eq. (2)), we use both the **GNN** and the **LLM** to make predictions on these selected nodes. For the **GNN**, we assign each node a pseudo label based on the class with the highest softmax probability. For the **LLM**, we directly prompt the model to output its predicted class using the text-based prompt templates described in Appendix F. We then compare the two sets of predictions: nodes where the GNN and LLM **agree** on the predicted label form the **agreement set**; nodes where the GNN and LLM **disagree** form the **disagreement set**.
>
> >**Q2: Simplified algorithm flowchart**
>
> Thank you for the helpful feedback. We have revised both the **flowchart** in **Figure 1** and the **pseudo-algorithm** in the **Appendix D** for greater clarity.
>
> > **Q3: Typo**
>
> We appreciate the reviewer’s attention to detail. All identified typos have been **corrected and highlighted in the revised PDF**.
>
> [1] Runjin Chen, Tong Zhao, Ajay Jaiswal, Neil Shah, and Zhangyang Wang. Llaga: Large language and graph assistant. In ICML, 2024.
>
> [2] Jiabin Tang, Yuhao Yang, Wei Wei, Lei Shi, Lixin Su, Suqi Cheng, Dawei Yin, and Chao Huang.
> Graphgpt: Graph instruction tuning for large language models. In SIGIR, 2024.
>
> [3] Xixi Wu, Yifei Shen, Fangzhou Ge, Caihua Shan, Yizhu Jiao, Xiangguo Sun, and Hong Cheng. When do llms help with node classification? a comprehensive analysis. In ICML, 2025.
>
> [4] Zhikai Chen, Haitao Mao, Hongzhi Wen, Haoyu Han, Wei Jin, Haiyang Zhang, Hui Liu, and Jiliang Tang. Label-free node classification on graphs with large language models (llms). In ICLR, 2024.
>
> [5] Zeang Sheng, Weiyang Guo, Yingxia Shao, Wentao Zhang, and Bin Cui. Llms are noisy oracles! llm-based noise-aware graph active learning for node classification. In KDD, 2025.
>
> [6] Jianan Zhao, Meng Qu, Chaozhuo Li, Hao Yan, Qian Liu, Rui Li, Xing Xie, and Jian Tang. Learning on large-scale text-attributed graphs via variational inference. In ICLR, 2023.
>
> [7] Duan, Keyu and Liu, Qian and Chua, Tat-Seng and Yan, Shuicheng and Ooi, Wei Tsang and Xie, Qizhe and He, Junxian. Simteg: A frustratingly simple approach improves textual graph learning. arXiv preprint arXiv:2308.02565.

---

> ### Author Response · Authors · 2025-11-26
> **Appreciation for Your Review and Awaited Feedback**
>
> Dear Reviewer EAHZ,
>
> We sincerely appreciate your time and effort in reviewing our manuscript.
>
> We have posted our responses to your feedback and a revised manuscript. With one week remaining in the discussion period, we would like to confirm whether our responses have effectively addressed your concerns.
>
> If you have any further questions or need clarification, please let us know and we're happy to address them promptly.
>
> Thank you again for your valuable feedback.
>
> Best regards,
> Authors

---

### Official Review · Reviewer_HTs2 · 2025-11-01

**Soundness:** 3
**Presentation:** 3
**Contribution:** 3
**Rating:** 2
**Confidence:** 5

**Summary:**

This paper studies few-shot semi-supervised node classification on text-attributed graphs (TAGs). It proposes GNN-as-Judge, a framework that combines a structure-aware GNN and a text-centric LLM to curate high-quality pseudo-labels and mitigate label noise. The method first selects a subset of unlabeled nodes using an influence-guided criterion, then exploits LLM–GNN agreement and disagreement with GNN preference scores to construct pseudo-labeled data. The LLM is fine-tuned with a weakly supervised objective that integrates instruction tuning on the agreement set and preference tuning on the disagreement set. Experiments on multiple benchmarks show consistent gains, especially under low-resource regimes.

**Strengths:**

Strengths:
1) GNN-as-Judge paradigm: Positioning the GNN as a judge complements the LLM’s lack of structural inductive bias, yielding more reliable pseudo-labels on TAGs.
2) Collaborative pseudo-labeling: Using both agreement and disagreement—filtered by GNN preference margins—captures easy and hard examples, enhancing learning signals beyond confidence-only self-training.
3) Weakly supervised fine-tuning: The combined instruction-tuning plus preference-tuning objective leverages informative but potentially noisy hard examples while reducing overfitting risk.
4) Empirical validation: The approach outperforms strong GNN and LLM-based baselines across 3/5/10-shot settings and shows promising cross-dataset zero-shot transfer, indicating practical value in low-label scenarios.

**Weaknesses:**

Weaknesses:
1) Gap Between Theory and Practice: The paper's theoretical proofs rely on simplified assumptions (like linear models and independent errors) that may not hold true in the actual, more complex experimental setup.
2) Unreliable GNN Judge: The method assumes the GNN is a trustworthy "judge" when it disagrees with the LLM. This assumption can fail on graphs with noisy structures or unusual patterns, leading the LLM astray with flawed guidance.
3) High Computational Cost: The multi-stage pipeline is computationally expensive, demanding significant time and memory. This raises practical concerns about its scalability to massive, real-world graphs.
4) Complex Hyperparameter Tuning: The framework introduces several new hyperparameters that need careful tuning for each dataset. This adds complexity and makes the method harder to apply effectively without extensive experimentation.

**Questions:**

n/a

---

> ### Author Response · Authors · 2025-11-19
> **Response Part 1.**
>
> We sincerely thank the reviewer for the thorough evaluation and for acknowledging the soundness of our method (Soundness: 3), the quality of our presentation (Presentation: 3), and the solid contribution of our work (Contribution: 3). We carefully address each concern below and have incorporated suggested improvements into our revised manuscript.
>
> > W1: Gap Between Theory and Practice
>
> We thank the reviewer for raising this concern and respectfully clarify our theoretical framework with additional empirical validation.
>
> **Regarding the Linear Assumption:** Our linear analysis follows standard practices in GNN theory, as adopted by prior theoretical works [1,2], which assume identity activation and weight matrices for analytical tractability. Importantly, this assumption directly corresponds to the practical GNN architecture Simple Graph Convolution (SGC) [3], which explicitly removes nonlinear activation functions between layers and collapses weight matrices into a single linear transformation. Wu et al. [3] demonstrate that SGC achieves competitive performance on node classification tasks and conclude that "the expressive power of GCNs originates primarily from the repeated graph propagation (which SGC preserves) rather than the nonlinear feature extraction (which it doesn't)." This finding validates that linear assumptions capture the essential mechanisms of graph learning. Moreover, our theoretical framework can be readily extended to other GNN architectures (such as GAT [4] and GraphSAGE [5]) by assigning different values to edge weights. Furthermore, **Table 2** demonstrates that our influence-guided selection substantially outperforms alternative strategies across all datasets, validating the practical effectiveness of our theoretical insights even when applied to nonlinear GNN architectures.
>
> **Regarding the Independence Assumption:** We acknowledge that our theoretical derivation assumes independent errors between models. We conducted comprehensive error correlation analysis across all datasets:
>
> | Dataset       | Pearson | Δ_L\|G | Δ_G\|L |
> | ------------- | ------- | ------ | ------ |
> | Cora          | 0.258   | 0.216  | 0.162  |
> | Citeseer      | 0.202   | 0.159  | 0.115  |
> | Pubmed        | -0.048  | 0.020  | 0.062  |
> | ogbn-arxiv    | 0.117   | 0.047  | 0.058  |
> | ogbn-products | 0.140   | 0.061  | 0.094  |
>
> **Pearson correlation** captures linear dependence between error indicators, while **Δ_L|G = P(LLM err | GNN err) - P(LLM err)** measures how much the LLM error rate increases given that the GNN errs (and vice versa for Δ_G|L). The consistently small values across all datasets indicate weak error correlation. The empirical results strongly support our framework: agreement-set accuracy consistently exceeds both individual model accuracies (**Table 5**).
>
> While we acknowledge that absolute independence is rarely satisfied in real-world scenarios, we provide more rigorous theoretical coverage for non-zero correlation through **Lemma E.3 (Appendix E)**, which establishes a sufficient condition for agreement-set superiority under bounded error dependence. Specifically, the lemma shows that as long as the Pearson correlation ρ satisfies ρ < ρ_max (a computable threshold depending on individual model accuracies and the number of classes), the agreement-set accuracy will exceed both individual models. Across all datasets, the observed correlations satisfy this condition, confirming that our theoretical guarantees hold even with weak correlation.
>
> We have included all detailed analyses in **Appendix E.2** (theoretical proofs), **Appendix G.1** (agreement patterns), and **Appendix H.5** (error correlation analysis) in the revised manuscript.

---

> ### Author Response · Authors · 2025-11-19
> **Response Part 2.**
>
> > W2: Unreliable GNN Judge
>
> We sincerely thank the reviewer for raising this critical concern about the reliability of the GNN as a judge. We provide extensive empirical analysis below demonstrating that our method is robust across different graph structures and can be adapted to noisy graphs.
>
> In our main experiments, we evaluate on five commonly used datasets with **varying properties and structures**. Our method does not blindly trust GNN judgments or use all GNN signals over LLM predictions. Instead, we employ a preference-score filtering mechanism that selectively incorporates GNN guidance only when it is likely to be reliable, effectively reducing noisy supervision. The key evidence is in Table 5's last column "Sel. Disagree Acc.", which shows accuracy after applying our filtering mechanism. This filtering yields substantial improvements: PubMed (31.55% → 40.03%, +8.48%), ogbn-arxiv (37.38% → 59.37%, +21.99%), and ogbn-products (37.24% → 52.29%, +15.05%). These gains demonstrate that our preference score filtering successfully identifies cases where the GNN provides more reliable guidance, thereby reducing noisy supervision risk.
>
> Following other reviewer's suggestion, we provide a comprehensive breakdown of error patterns in the disagreement set, both *before* and *after* preference-score filtering:
>
> **Before Filtering**
>
> | Dataset      | GNN✓ LLM✗ (Beneficial) | LLM✓ GNN✗ (Harmful) | Both✗ (Harmful) |
> | ------------ | ---------------------- | ------------------- | --------------- |
> | **Cora**     | 321 (61.7%)            | 108 (20.8%)         | 91 (17.5%)      |
> | **Citeseer** | 409 (66.6%)            | 121 (19.7%)         | 84 (13.7%)      |
> | **PubMed**   | 159 (31.5%)            | 340 (67.5%)         | 5 (1.0%)        |
> | **Arxiv**    | 332 (37.4%)            | 251 (28.3%)         | 305 (34.3%)     |
> | **Products** | 273 (37.2%)            | 295 (40.3%)         | 165 (22.5%)     |
>
> **After Filtering**
>
> | Dataset      | GNN✓ LLM✗ (Beneficial) | LLM✓ GNN✗ (Harmful) | Both✗ (Harmful) |
> | ------------ | ---------------------- | ------------------- | --------------- |
> | **Cora**     | 101 (82.1%)            | 13 (10.6%)          | 9 (7.3%)        |
> | **Citeseer** | 172 (72.3%)            | 49 (20.6%)          | 17 (7.1%)       |
> | **PubMed**   | 80 (40.0%)             | 117 (58.5%)         | 3 (1.5%)        |
> | **Arxiv**    | 122 (59.5%)            | 44 (21.5%)          | 39 (19.0%)      |
> | **Products** | 68 (52.3%)             | 45 (34.6%)          | 17 (13.1%)      |
>
> The preference-score filtering significantly reduces cases where the LLM is correct but the GNN is wrong (harmful supervision). This pattern holds across all datasets, confirming that our filtering strategy effectively mitigates noisy information from the GNN judge. We have added this comprehensive error analysis to **Appendix G.5** in the revised manuscript.
>
> While our paper does not focus specifically on noisy structures, we emphasize that our framework is plug-and-play. The GNN backbone can be easily replaced with any architecture tailored for scenarios with noisy structures or unusual patterns (e.g., **robust GNNs designed for noisy graphs**), making our approach adaptable to challenging graph conditions without requiring fundamental changes to the overall framework.
>
> > W3: High Computational Cost
>
> We thank the reviewer for raising this concern. We included comprehensive time and memory analysis in the original manuscript (**Figure 5 in Section 4.6 and Appendix H.6**). Our results show that we are 2.5-3× faster than comparable LLM baselines while achieving higher accuracy.
>
>  Since we utilize influence scores to select only the most reliable nodes for annotation, the pseudo-labeling inference time does not scale up even for large graphs. The number of nodes requiring LLM inference and training remains bounded. We utilize LoRA as our fine-tuning strategy, which dramatically reduces both training time and memory consumption compared to full fine-tuning. A more detailed breakdown is provided in **Appendix H.8**.
>
> > W4: Complex Hyperparameter Tuning
>
> We thank the reviewer for bringing up this concern about hyperparameter tuning. Our sensitivity analysis (**Section 4.6 and Appendix H.1**) demonstrates that our method exhibits remarkable robustness to hyperparameter choices, making it highly practical even without extensive tuning. Critically, we use **identical hyperparameters across all 5 datasets** and still achieve superior performance compared to baseline methods. This validates our framework's effectiveness without requiring precise dataset-specific tuning. The robustness is particularly valuable for practical applications where validation data may be scarce or unavailable.
>
> We hope these clarifications address the reviewer's concerns. We are happy to provide additional experiments or analyses if needed.

---

> > ### Author Response · Authors · 2025-11-27
> > **References**
> >
> > [1] Huang, K., & Zitnik, M. (2020). Graph meta learning via local subgraphs. NeurIPS.
> >
> > [2] Xu, K., Hu, W., Leskovec, J., & Jegelka, S. (2018). How powerful are graph neural networks? ICLR.
> >
> > [3] Wu, F., Souza, A., Zhang, T., Fifty, C., Yu, T., & Weinberger, K. (2019). Simplifying graph convolutional networks. ICML.
> >
> > [4] Veličković, P., Cucurull, G., Casanova, A., Romero, A., Liò, P., & Bengio, Y. (2018). Graph attention networks. ICLR.
> >
> > [5] Hamilton, W., Ying, Z., & Leskovec, J. (2017). Inductive representation learning on large graphs. NeurIPS.

---

> ### Author Response · Authors · 2025-11-27
> **Appreciation for Your Review and Awaited Feedback**
>
> Dear Reviewer HTs2,
>
> We sincerely appreciate your time and effort in reviewing our manuscript.
>
> We have posted our responses to your feedback and a revised manuscript. With one week remaining in the discussion period, we would like to confirm whether our responses have effectively addressed your concerns.
>
> If you have any further questions or need clarification, please let us know and we're happy to address them promptly.
>
> Thank you again for your valuable feedback.
>
> Best regards,
> Authors

---

### Author Response · Authors · 2025-11-25
**Meta Response to All Reviewers and Revision Note**

Dear PCs, SACs, ACs, and Reviewers,

We sincerely thank all reviewers for their constructive and insightful feedback. Below is a consolidated summary of the major revisions.  (All technical changes appear in **purple** in the revised PDF.)

> **1. Theoretical & Methodological Clarifications (HTs2, EAHZ, nL51)**
- Added comprehensive **error correlation analyses** across all datasets, confirming weak dependence between GNN and LLM errors (**Appendix H.5**).
- Added **Lemma E.3**, formally establishing agreement-set superiority under bounded error correlation.

> **2. Agreement/Disagreement Analysis & Preference Filtering (HTs2, nL51)**
- Added detailed breakdowns of agreement and disagreement cases before and after preference-score filtering, with full error analysis (**Appendix G.5**).

> **3. Influence-Guided Node Selection & Bias Evaluation (nL51)**
- Added distribution-level comparisons (KL/JS divergence, total variation) showing that selected nodes do not overrepresent majority classes (**Appendix G.2**).
- Added class-specific improvement analyses for both influence selection and preference tuning (**Appendix G.7**).
- Added additional experiments evaluating performance under imbalanced node distributions (**Appendix H.10**).
- Added class-level benefit analyses illustrating how different classes gain from influence-guided selection and preference tuning (**Appendix H.11**).

> **4. Hyperparameter Sensitivity (HTs2, 19Gq)**
- Added an expanded table showing set size vs. accuracy vs. τ, highlighting stable performance across thresholds (**Appendix G.6**).

> **5. Training Time & Computational Efficiency (HTs2, 19Gq)**
- Added a full runtime decomposition covering influence computation, GNN training, LLM fine-tuning, and LLM inference (**Appendix H.8**).

> **6. Architecture Robustness & Additional Experiments (19Gq)**
- Added new results demonstrating robustness when using GAT and GraphSAGE as alternative GNN judges (**Appendix H.9**).

> **7. Flowchart & Algorithm Presentation (EAHZ)**
- Added a concise and clearer definition of the agreement and disagreement sets (**Sec. 3**).
- Streamlined the algorithm flowchart and reduced redundant notation in **Figure 1**.
- Expanded and contextualized the related work discussion (**Section 2**).
- Added a concise and clarified pseudo-algorithm (**Appendix D**).
- Refined the main theoretical statements for clarity and brevity (**Sec. 3**).
- Corrected the “citepseer” typo and several minor inconsistencies.

Thank you again for your thoughtful feedback, which has greatly strengthened the paper.

---

### Meta-Review · Area_Chair_Ypkb · 2026-01-06

**Summary:**

This paper proposes GNN-as-Judge, a framework that integrates the complementary strengths of large language models and graph neural networks. The core idea is to employ a GNN as a structural judge to guide pseudo-label generation and selection for the LLM by partitioning structurally influential unlabeled nodes into agreement and disagreement sets based on prediction consistency. A weakly supervised fine-tuning strategy that combines instruction and preference tuning is further introduced. Extensive experiments on multiple text-attributed graph benchmarks demonstrate consistent improvements over state-of-the-art baselines. All reviewers find the idea compelling and well motivated, and the rebuttal adequately addresses their concerns. Thus, I recommend accepting this paper.

**Reviewer Concerns:**

The rebuttal adequately addressed the reviewers’ main concerns. In particular, the authors clarified the motivation and design of the GNN-as-Judge framework, provided additional explanations of the agreement–disagreement mechanism and bias in influence-guided node selection. As a result, no major reviewer concerns remain outstanding.

**Reviewer Scores:**

The authors’ responses clarified methodological details and experiments. The reviewer would likely have a more positive assessment and might slightly increase their score.

---

### Decision · Program_Chairs · 2026-01-26

Accept (Poster)